# Molecular recognition by multiple metal coordination inside wavy-stacked macrocycles

Takashi Nakamura [1], Yuya Kaneko[1], Eiji Nishibori[1] & Tatsuya Nabeshima [1]

Most biological and synthetic receptors for small organic molecules employ a combination of relatively weak intermolecular interactions such as hydrogen bonds. A host compound that utilizes stronger yet reversible bonding in a synergistic manner could realize precise recognition, but the regulation and spatial arrangement of such reactive interaction moieties have been a challenge. Here, we show a multinuclear zinc complex synthesized from a macrocyclic ligand hexapap, which inwardly arranges labile metal coordination sites for external molecules. The metallomacrocycle forms a unique wavy-stacked structure upon binding a suitable length of dicarboxylic acids via multipoint coordination bonding. The saddle-shaped deformation and dimerization realize the differentiation of the interaction moieties, and change of guest-binding modes at specific metal coordination sites among the many present have been achieved utilizing acid/base as external stimuli.

---

[1] Graduate School of Pure and Applied Sciences and Tsukuba Research Center for Interdisciplinary Materials Science (TIMS), University of Tsukuba, 1-1-1 Tennodai, Ibaraki, Tsukuba 305-8571, Japan. Correspondence and requests for materials should be addressed to T.N. (email: nabesima@chem.tsukuba.ac.jp)

Precise recognition of small molecules plays a vital role in nature. It serves as a basis for sophisticated functions such as signal transduction. Molecular recognition in biological systems[1] is usually realized by the combination of relatively weak intermolecular interactions, such as hydrogen bonds (typically 10–40 kJ/mol)[2], aromatic–aromatic interactions (~5 kJ/mol)[3], and van der Waals interactions (~1 kJ/mol)[2]. Many artificial synthetic receptors[4–6] utilizing these interactions have already been developed, but they are still not optimized in terms of specificity. To achieve sophisticated recognition events, it is required to properly and three-dimensionally incorporate multiple interaction moieties into the molecular-binding sites, but chemically synthetic receptors are no match against biological counterparts in that respect.

In this context, utilization of stronger and more directional interactions in a synergistic manner could create a sophisticated artificial host. That is, a host that utilizes multiple coordination bonds. Coordination bonds between metal atoms and Lewis bases are categorized to be stronger than hydrogen bonds, but weaker than typical covalent bonds (~500 kJ/mol, for a C–C bond)[2]. As an example using zinc, the binding enthalpy of the reaction $[Zn(H_2O)_6]^{2+} \rightarrow [Zn(H_2O)_5]^{2+} + (H_2O)$ was calculated to be ~120 kJ/mol[7]. Although the coordination bonds are relatively strong, they are labile and reversible enough to be used for molecular recognition, which can be seen from the very fast exchange reaction of a water on the $[Zn(H_2O)_6]^{2+}$ with the lifetime on the order of $10^{-7}$ s[8]. To utilize multiple coordination bonds in the molecular recognition events[9, 10], labile coordination sites on the metal centers should be spatially arranged in the binding pocket. External small molecules bind to multiple metals in place of the exchangeable ligands.

In this study, we create a rigid macrocyclic ligand hexapap $H_6\mathbf{1}$, and embed the metals in its cavity[11–16] with their labile coordination sites directed toward the center of the pore.

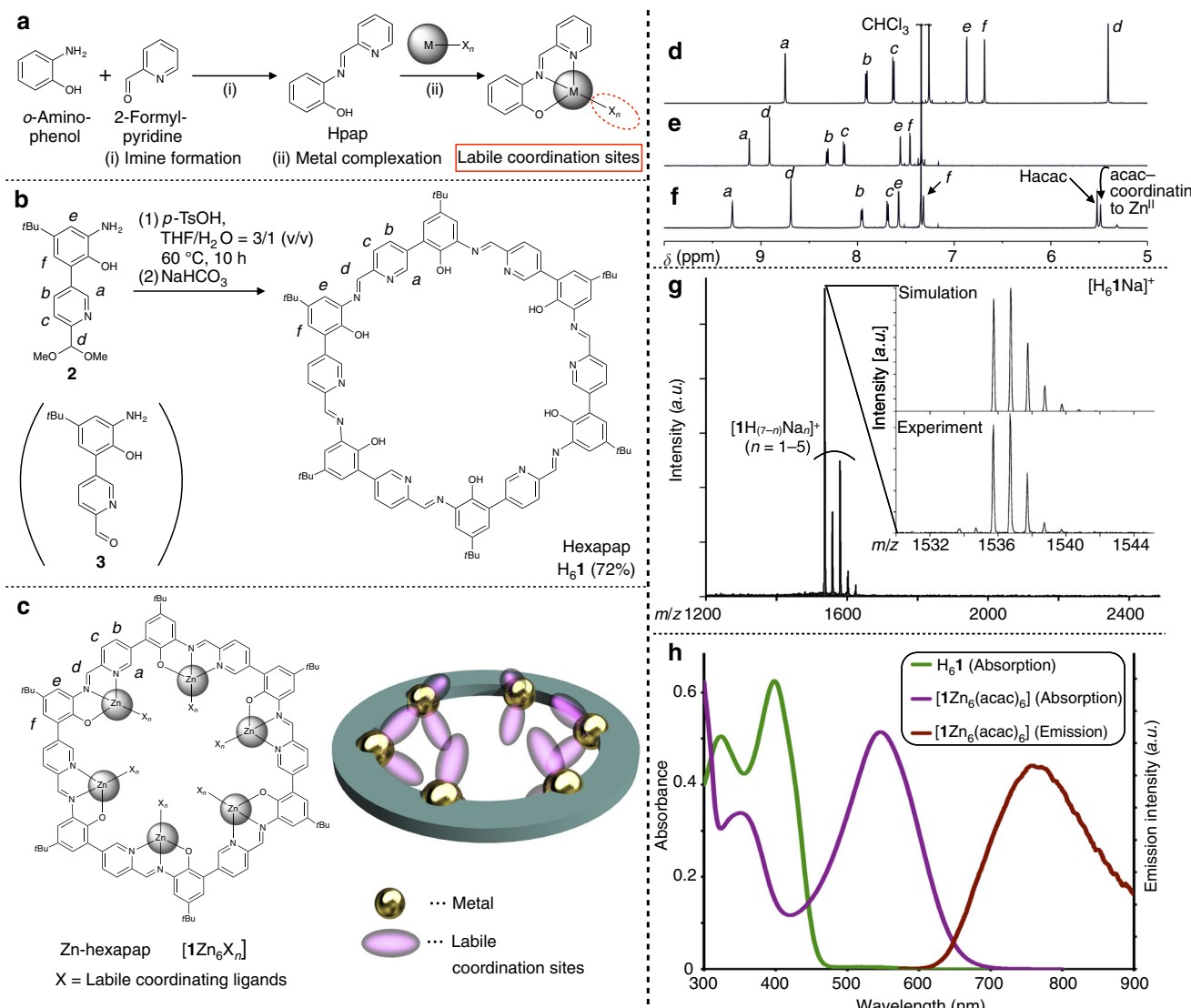

**Fig. 1** Synthesis and characterization of hexapap ligand and Zn-hexapap. **a** Formation of an *N,N,O*-type tridentate ligand Hpap and its metal complex. **b** Synthesis of hexapap $H_6\mathbf{1}$ from the bifunctional monomer **2** by a one-pot reaction. **c** Chemical structure and schematic representation of a metallomacrocycle, Zn-hexapap [$\mathbf{1}Zn_6X_n$], with inwardly arranged coordination sites. **d–f** [1]H NMR spectra (600 MHz, 298 K). See **b**, **c** for the assignment of NMR signals. **d** 2 (CDCl$_3$). **e** $H_6\mathbf{1}$ (CDCl$_3$/CD$_3$OD = 10/1 (v/v)). **f** [$\mathbf{1}Zn_6(acac)_6$] (CDCl$_3$/CD$_3$OD = 10/1 (v/v)). **g** A MALDI TOF mass spectrum of $H_6\mathbf{1}$ (positive, matrix: 2,5-dihydroxybenzoic acid). The simulated and observed isotope patterns of [$H_6\mathbf{1}Na$]$^+$ are shown in the inset. **h** Absorbance spectra of $H_6\mathbf{1}$ (*green*) and [$\mathbf{1}Zn_6(acac)_6$] (*purple*) and emission spectrum of [$\mathbf{1}Zn_6(acac)_6$] (*red*, $\lambda_{ex}$ = 546 nm) (5 μM, CHCl$_3$/CH$_3$OH = 10/1 (v/v), 298 K, *l* = 1.0 cm)

The complexation of $H_6\mathbf{1}$ with $Zn^{II}$ produces a hexanuclear complex, Zn-hexapap [$\mathbf{1}Zn_6X_n$] (X = exchangeable labile ligands). Interestingly, the dicarboxylic acids with suitable chain lengths induce the formation of a uniquely-shaped wavy-stacked dimer of the Zn-hexapap via multiple coordination bonds between the carboxylate groups and Zn. Although the monomeric Zn-hexapap [$\mathbf{1}Zn_6X_n$] has six chemically equivalent metal centers, the saddle-shaped deformation and dimerization of the macrocycles realize the differentiation of the Zn(pap) units. This desymmetrized dimeric macrocycle achieves the regulation and change of the guest-binding modes at specific metal coordination sites among the many available utilizing acid/base as external stimuli.

## Results

**Synthesis of hexapap ligand and Zn-hexapap complex.** $H_6\mathbf{1}$ possesses six inward Hpap (2-[(pyridin-2-ylmethylene)amino] phenol) chelate-binding units[17, 18] (Fig. 1a). Pap⁻ is a negatively-charged tridentate ligand. Upon binding of a metal, labile coordination sites not occupied by pap⁻ are available for guest binding. Meanwhile, the tridentate chelation of the metal is strong enough to prevent its removal by external guests.

Reversible imine bonds are often utilized for the construction of thermodynamically stable target products[19–21], and most of them were constructed by mixing aldehyde and amine building blocks[22–24]. Here, $H_6\mathbf{1}$ was synthesized from a bifunctional monomer $\mathbf{3}$ possessing both o-aminophenol and 2-formylpyridine moieties (Fig. 1b). The compound $\mathbf{2}$, a derivative of $\mathbf{3}$, was designed whose formyl group was protected by dimethyl acetal to prevent spontaneous self-oligomerization[25]. $\mathbf{2}$ was prepared as an isolable cyclization precursor (¹H nuclear magnetic resonance (NMR), Fig. 1d) (see Supplementary Figs. 1–10 for the synthesis of $\mathbf{2}$ and its synthetic intermediates). The angle between the formyl group and the amino group installed into $\mathbf{3}$ was about 120°. This geometrical feature realized the high-yield synthesis of the hexagonal macrocycle hexapap $H_6\mathbf{1}$. That is, an aqueous acid-catalyzed reaction (p-toluene sulfonic acid, tetrahydrofuran (THF)/$H_2O$ = 3/1 (v/v)) facilitated deprotection of the formyl group of $\mathbf{2}$ and imine-bond formation, and produced $H_6\mathbf{1}$ in 72% yield. $H_6\mathbf{1}$ was obtained as a yellow precipitate in this reaction. The ¹H NMR spectrum of the solid indicated the formation of $H_6\mathbf{1}$ as a single, pure product, and its symmetric time-averaged structure on the NMR time scale (~msec) (Fig. 1e and Supplementary Fig. 11). The composition and purity of $H_6\mathbf{1}$ as a cyclic hexamer was further supported by MALDI TOF mass (Fig. 1g), infrared (IR), and elemental analysis (see Methods).

The reaction of $H_6\mathbf{1}$ and $Zn(acac)_2$ (acac⁻ = acetylacetonate) led to the formation of Zn-hexapap [$\mathbf{1}Zn_6(acac)_6$]. $Zn^{II}$ centers of Zn-hexapap adopted a five-coordinate trigonal-bipyramidal geometry. Two labile coordination sites per Zn atom are available, which are directed upwardly inward and downwardly inward (Fig. 1c). Changes in the chemical shifts of $\mathbf{1}^{6-}$ were observed in the ¹H NMR spectrum upon the complexation of Zn, and [$\mathbf{1}Zn_6(acac)_6$] retained its time-averaged sixfold symmetry (Fig. 1f). The acac⁻ ligands coordinating to the Zn from the inside of the macrocycle were clearly discerned in the ¹H NMR spectrum. Complexation of Zn to pap⁻ was further characterized by the absorption and emission at the 546 nm and 762 nm, respectively (Fig. 1h). The absorption maximum (546 nm) of [$\mathbf{1}Zn_6(acac)_6$] did not change at the concentrations of 1–100 μM, and ¹H NMR signals of the complex were also not shifted depending on the concentrations (0.15–1.2 mM) (see Supplementary Figs. 12 and 13). These results indicated that stacking of the macrocycles did not occur under these conditions.

**Wavy-staked dimer of Zn-hexapap with dicarboxylic acids.** Carboxylic acids were investigated as guest molecules in this study, for their ubiquity in nature and coordination ability of

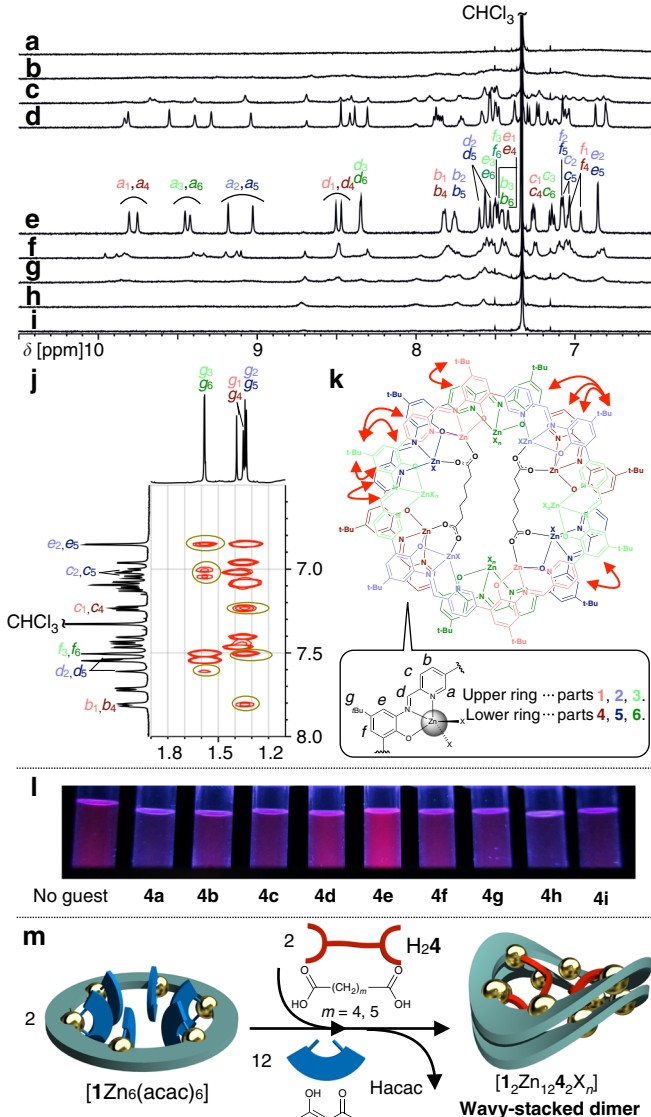

**Fig. 2** Binding of dicarboxylic acids by Zn-hexapap and the formation of the wavy-stacked dimer. **a–i** Interaction of dicarboxylic acids $H_2\mathbf{4a}$–$H_2\mathbf{4i}$ and Zn-hexapap [$\mathbf{1}Zn_6(acac)_6$].(¹H NMR, 600 MHz, CDCl₃/CD₃OD = 10/1 (v/v), 298 K, [$\mathbf{1}Zn_6(acac)_6$] = 2.5 mM). **a** Malonic acid $H_2\mathbf{4a}$ (m = 1). m indicates the number of methylene groups between the two carboxylic groups. **b** Succinic acid $H_2\mathbf{4b}$ (m = 2). **c** Glutaric acid $H_2\mathbf{4c}$ (m = 3). **d** Adipic acid $H_2\mathbf{4d}$ (m = 4). **e** Pimelic acid $H_2\mathbf{4e}$ (m = 5). See **k** for assignment of the signals. **f** Suberic acid $H_2\mathbf{4f}$ (m = 6). **g** Azelaic acid $H_2\mathbf{4g}$ (m = 7). **h** Sebacic acid $H_2\mathbf{4h}$ (m = 8). **i** Dodecanedioic acid $H_2\mathbf{4i}$ (m = 10). **j** ¹H–¹H ROESY (rotating-frame Overhauser effect spectroscopy) NMR spectrum of the complex with two pimelates $\mathbf{4e}^{2-}$, [$\mathbf{1}_2Zn_{12}\mathbf{4e}_2X_n$] (X = labile coordinating ligand) (600 MHz, CDCl₃/CD₃OD = 10/1 (v/v), 323 K). Yellow circles indicate ROE cross peaks between the top and bottom macrocycles. **k** Chemical structure of [$\mathbf{1}_2Zn_{12}\mathbf{4e}_2X_n$]. Red arrows indicate the pairs of ¹H–¹H between which the ROE cross peaks were observed (see Supplementary Fig. 18). See also Fig. 3d for the crystal structure of [$\mathbf{1}_2Zn_{12}\mathbf{4e}_2(H_2O)_4Cl_8$] colored in the same manner. **l** Emissions from Zn-hexapap during UV irradiation (365 nm) upon binding of a series of dicarboxylic acids $H_2\mathbf{4a}$–$H_2\mathbf{4i}$ (10 μM, CHCl₃/CH₃OH = 10/1 (v/v), 298 K). **m** A schematic representation of the recognition of dicarboxylic acids in the cavity of the wavy-stacked dimer of the Zn-hexapap

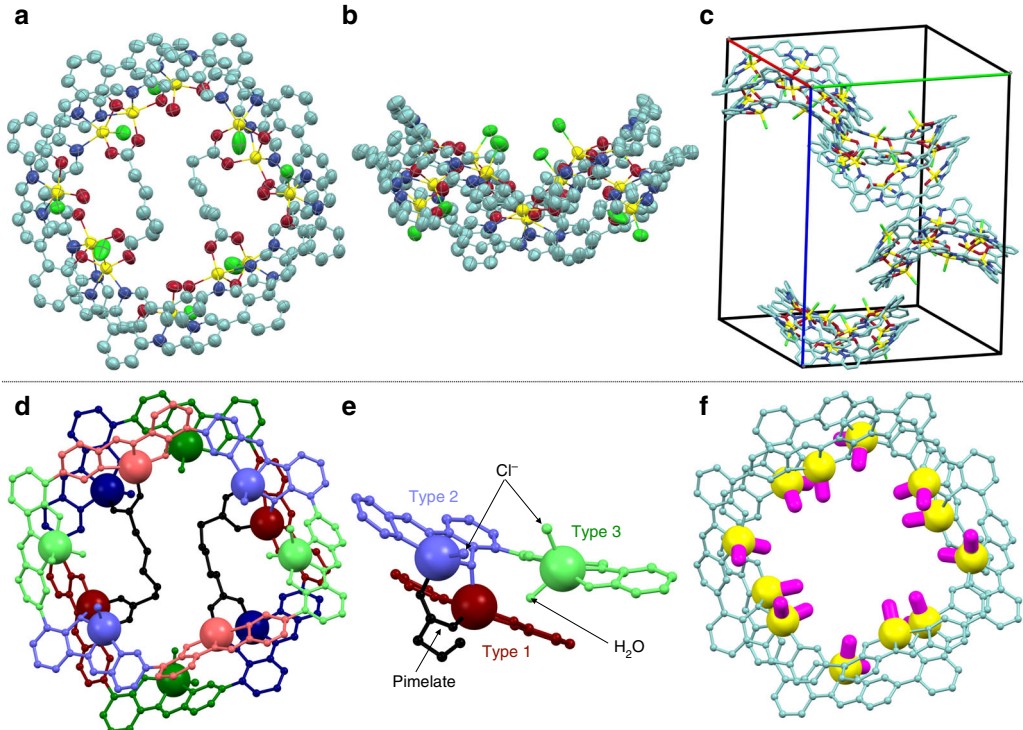

**Fig. 3** Structure of [$\mathbf{1}_2Zn_{12}\mathbf{4e}_2(H_2O)_4Cl_8$] determined by X-ray crystallography. Solvents, hydrogens, and *t*Bu groups are omitted for clarity. One disorder pattern of $\mathbf{4e}^{2-}$ is shown. **a, b** An ellipsoidal model (30% probability). **c** Packing in the crystal (a stick model). For **a–c**, the atoms are colored according to the elements: C, *light green*; N, *blue*; O, *red*; Zn, *yellow*; Cl, *green*. **d** A ball-and-stick model. **e** Three different coordination modes around the Zn centers. For **d** and **e**, the atoms are colored to show the pseudo $C_2$ symmetry of the entire structure (see also Fig. 2k). The Zn atoms are described in a space-filling model. **f** 20 coordination bonds (*magenta*) that were not occupied by the *N,N,O*–chelating moieties of $\mathbf{1}^{6-}$. Zn, *yellow*; non-metal atoms, *light green*

their carboxylate groups to labile coordinating sites of metal centers. The recognition experiments of aliphatic dicarboxylic acids HOOC-(CH$_2$)$_m$-COOH ($m = 1$–8, 10) by Zn-hexapap [$\mathbf{1}Zn_6(acac)_6$] are shown in Fig. 2. A series of $^1$H NMR spectra (Fig. 2a–i) suggest that only adipic acid H$_2\mathbf{4d}$ ($m = 4$) and pimelic acid H$_2\mathbf{4e}$ ($m = 5$) led to the formation of a single species. In other words, a clear dependence of the molecular length was observed in this coordination-driven recognition event. The formation of a certain host–guest complex was also supported by the change in emission, where the samples in which H$_2\mathbf{4d}$ or H$_2\mathbf{4e}$ was mixed with [$\mathbf{1}Zn_6(acac)_6$] showed a stronger red emission than the other dicarboxylic acids (Fig. 2l, Supplementary Fig. 14, and Supplementary Table 1). The $^1$H NMR spectrum of the host–guest complex with H$_2\mathbf{4e}$ suggested that its entire structure was desymmetrized and six different pap$^-$ moieties were present in the structure (Fig. 2e). The ESI-TOF (electrospray ionization time-of-flight) mass spectrum of the sample with H$_2\mathbf{4e}$ indicated that the dimer of Zn-hexapap [$\mathbf{1}Zn_6X_n$] was formed with two pimelate $\mathbf{4e}^{2-}$ molecules, that is, [$\mathbf{1}_2Zn_{12}\mathbf{4e}_2X_n$] (see Supplementary Fig. 15).

A single crystal suitable for X-ray diffraction analysis was obtained by the slow diffusion of acetone vapor into a 1,1,2,2-tetrachloroethane/MeOH solution of [$\mathbf{1}_2Zn_{12}\mathbf{4e}_2X_n$] (Fig. 3 and Supplementary Fig. 16). Interestingly, it was found that the macrocyclic framework of hexapap $\mathbf{1}^{6-}$ uniquely warped, while it was tightly stacked around the total circumference of the macrocycle to form a dimeric structure (Fig. 3a, b). Each wavy-stacked dimer was not aligned on a concentric axis in the crystal, but found to be slip-stacked on a $4_1$ helical axis (Fig. 3c). As expected from the ESI mass measurement, two molecules of the pimelates $\mathbf{4e}^{2-}$ were captured inside the dimer of the Zn-hexapap. $\mathbf{4e}^{2-}$ was recognized though multipoint coordination bonding with the metallomacrocycles. The two terminal

carboxylate groups of $\mathbf{4e}^{2-}$ both bridged two Zn atoms. One of the Zn atoms belonged to the top macrocycle, while the other to the bottom one (Fig. 3a, d). All the Zn centers adopted a five-coordinate trigonal-bipyramidal geometry, but they can be categorized into three types in terms of the coordinating ligands at the inner exchangeable coordination sites (Fig. 3e). The first type of Zn (depicted in red in Fig. 3d, e) was coordinated by a carboxylate oxygen atom of $\mathbf{4e}^{2-}$ and a phenoxy oxygen atom of the pap$^-$. The coordination bond between the Zn (type1) and phenoxy oxygen bridged the two Zn-hexapap macrocycles. The second type (depicted in blue) was bound by another carboxylate oxygen atom of $\mathbf{4e}^{2-}$ and a Cl$^-$. The third type (depicted in green) was bound by a Cl$^-$ and a water. Thus, Zn (type 3) was free from the guest molecule $\mathbf{4e}^{2-}$ (Cl$^-$ probably derived from the decomposition of 1,1,2,2-tetrachloroethane used as the crystallization solvent. We assumed that the slow generation of Cl$^-$ helped to grow single crystals with good qualities). From the structural analysis described above, the following three main factors are considered to be the driving forces for the formation of the dimeric structure [$\mathbf{1}_2Zn_{12}\mathbf{4e}_2X_n$]: (i) Inter-macrocycle coordination bonds between the Zn (type 1) and phenoxy oxygen; (ii) Coordination of the carboxylate groups of $\mathbf{4e}^{2-}$ bridging two Zn atoms (types 1 and 2); and iii) π–π stacking between the hexapap aromatic frameworks.

The top and bottom macrocycles were in a different environment as the result of the binding of $\mathbf{4e}^{2-}$. The overall structure of [$\mathbf{1}_2Zn_{12}\mathbf{4e}_2(H_2O)_4Cl_8$] had a pseudo $C_2$ symmetry (Fig. 3d), which is consistent with the $^1$H NMR observation in the solution state. All the $^1$H NMR signals of [$\mathbf{1}_2Zn_{12}\mathbf{4e}_2X_n$] were successfully assigned based on the $^1$H-$^1$H COSY and $^1$H-$^1$H ROESY measurements (Fig. 2e, j, Supplementary Figs. 17, 18). Several characteristic ROE crosspeaks confirmed that the wavy

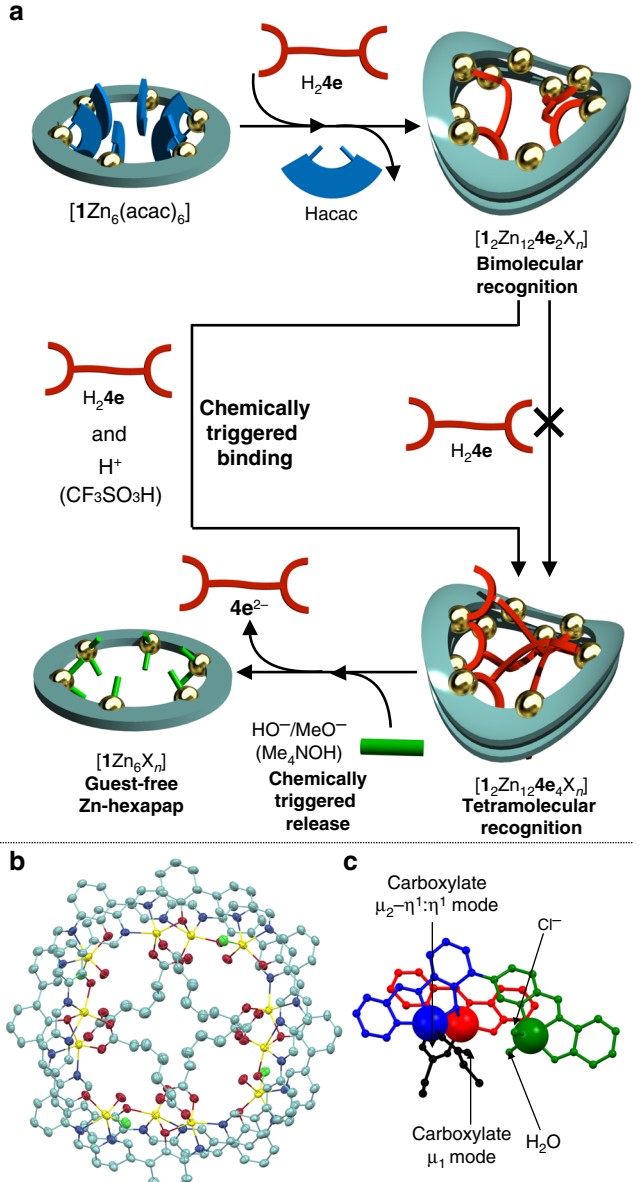

while 20 coordination sites in total were unoccupied by the $N,N$, $O$-chelating moieties of the hexapap ligand and arranged inward (Fig. 3f). Despite possessing many possible coordinating sites, the addition of more than two molar amounts of the pimelic acids $H_2 4e$ against the wavy-stacked dimer $[1_2Zn_{12}X_n]$ did not change a binding mode, but the host–guest complex stably existed in a bimolecular recognition mode $[1_2Zn_{12}4e_2X_n]$ (see Supplementary Fig. 19). Interestingly, however, an acid stimulus ($CF_3SO_3H$) triggered further incorporation of two more $4e^{2-}$'s, and led to a tetramolecular recognition mode $[1_2Zn_{12}4e_4X_n]$ (Fig. 4a, Supplementary Fig. 20). The conversion and the resulting complex were examined by $^1H$ NMR (Supplementary Figs. 20–22), ESI-TOF mass (Supplementary Fig. 23), and X-ray crystallographic analysis (*vide infra*). Three different $[Zn(pap)X_n]$ moieties were observed in the $^1H$ NMR spectrum of $[1_2Zn_{12}4e_4X_n]$, which suggested a change in the binding mode from $[1_2Zn_{12}4e_2X_n]$ ($C_2$ symmetry, six different $[Zn(pap)X_n]$ units) (Supplementary Fig. 20). The molecular structure of $[1_2Zn_{12}4e_4(H_2O)_4Cl_4]$ was revealed by the X-ray crystallographic analysis (Fig. 4b, c and Supplementary Fig. 24) (recrystallized from 1,1,2,2-tetrachloroethane/MeOH/acetone). The curvature of the wavy dimeric frameworks of the $[1_2Zn_{12}X_n]$ was slightly less bent for the complex with four $4e^{2-}$'s than the one with two $4e^{2-}$'s, although the frameworks were basically the same for the two complexes (see Supplementary Fig. 25). The binding mode of the $4e^{2-}$ in the structure of $[1_2Zn_{12}4e_4(H_2O)_4Cl_4]$ showed an interesting difference compared to the complex with two $4e^{2-}$'s (Fig. 4c). One carboxylate group of $4e^{2-}$ bridged two Zn atoms in a $\mu_2–\eta^1:\eta^1$ coordination mode, while the carboxylate group at the other end of $4e^{2-}$ was bound to Zn in a monodentate mode ($\mu_1$ mode). In terms of the Zn centers, there are three types of $[Zn(pap)X_n]$ moieties as in the case with the complex with two $4e^{2-}$'s. The entire structure of $[1_2Zn_{12}4e_4(H_2O)_4Cl_4]$ had an $S_4$ symmetry, which is consistent with the $^1H$ NMR observation in solution (see Supplementary Fig. 20).

The acid-triggered binding of $H_2 4e$ was explained by the protonation and following release of $HO^-$ or $MeO^-$ coordinating to the $[Zn(pap)]$ units. In solution, water and/or methanol molecules were bound to the labile coordination sites of $[1_2Zn_{12}4e_2X_n]$ (mainly water-bound complexes were observed in ESI-TOF mass and X-ray measurements). It is considered that the water/methanol molecules at the inner labile coordination sites initially exist as deprotonated $HO^-/MeO^-$ forms. The weak carboxylic acid $H_2 4e$ failed to protonate those ligands, but only the strong acid $CF_3SO_3H$ was able to protonate them. This is consistent with the fact that the $pK_a$ values of $H_2O$ bound to a Zn complex are in the range of 6–9, i.e., neutral pH[9]. This protonation weakened the coordination strength of the $H_2O/MeOH$ coordinating to Zn (type 2) in $[1_2Zn_{12}4e_2X_n]$, resulting in the replacement of them with the additional pimelates $4e^{2-}$ to produce $[1_2Zn_{12}4e_4X_n]$. Furthermore, the strategy to use acid/base stimuli to control the coordination strength can be applied to the release of guest molecules. That is, the addition of the $Me_4NOH$ base to the solution of $[1_2Zn_{12}4e_4X_n]$ released the guest $4e^{2-}$ and produced the guest-free Zn-hexapap $[1Zn_6X_n]$ (Fig. 4a, Supplementary Fig. 20). Here, $HO^-$ or $MeO^-$ worked as relatively strong ligands under basic conditions, and dissociated the host–guest complex into $[1Zn_6X_n]$, whose inner coordinating sites were occupied by $HO^-/MeO^-$. To summarize, the unique property of the wavy-stacked macrocycles Zn-hexapap to express multiple modes of molecular recognition via coordination bonds was demonstrated.

**Fig. 4** Control of guest binding via multiple metal coordination by acid/base stimuli. **a** A schematic representation. **b, c** The molecular structure of the complex with four pimelates $4e^{2-}$, $[1_2Zn_{12}4e_4(H_2O)_4Cl_4]$, determined by single-crystal X-ray crystallography. See Fig. 3 for the detailed description rules. **b** An ellipsoidal model (30% probability). **c** Coordination geometries of $4e^{2-}$ to the Zn centers

saddle-shaped structure was fixed on the NMR timescale and was essentially the same as the one determined by X-ray crystallography (Fig. 2k).

The wavy shape was considered to be the result of the clipping by the intermacrocyclic coordination bonds. That is, the difference in lengths between the clipped parts caused the distortion. Looking at the right-half structure of $[1_2Zn_{12}4e_2X_{12}]$ in Fig. 2k, the upper ring (light colors) has the part 3 between the clipped parts 2 and 1, but the clipped parts of the other ring (dense colors), parts 4 and 5, are adjacent to each other (see Fig. 2k for the part numbers).

**Regulation of guest binding at specific coordination sites.** In the wavy-stacked dimer of the Zn-hexapap, four coordination bonds were used to connect the two stacked macrocycles,

## Discussion

To summarize, we have designed and synthesized a hexapap ligand $H_6 1$, and its Zn complex Zn-hexapap $[1Zn_6X_n]$ that

has the inner cavity in which labile coordination bonds are spatially arranged. Zn-hexapap recognized the dicarboxylic acid $4e^{2-}$ though multiple coordination bonding to form the unique wavy-stacked dimeric structure. Furthermore, the clear control and change of the binding modes of the guest was achieved, although the metallocyclic dimer $[1_2Zn_{12}X_n]$ possesses as many as 20 available labile coordination sites. Thus, hexapap is shown to be an artificial host molecule that achieves the binding and the control of small molecules via multiple coordination bonds in solution. Metal complexes of the hexapap are promising platforms for selective molecular sensors[6] as well as for allosteric catalysts[26] that specifically interact with a target substrate via cooperative coordination, and such kinds of applications are now being investigated.

## Methods

**General**. Unless otherwise noted, solvents and reagents were purchased from TCI Co., Ltd., Wako Pure Chemical Industries, Ltd., Kanto Chemical Co., Inc., Nacalai Tesque, Inc. or Sigma-Aldrich Co., and used without further purification. THF was purified by Nikko Hansen Ultimate Solvent System 3S-TCN 1.

Measurements were performed at 298 K unless otherwise noted. $^1H$, $^{13}C$, and other 2D NMR spectra were recorded on a Bruker AVANCE III-600 (600 MHz) spectrometer or a Bruker AVANCE III-400 (400 MHz) spectrometer. Tetramethylsilane was used as an internal standard ($\delta$ 0.00 ppm) for $^1H$ and $^{13}C$ NMR measurements when CDCl$_3$ or a mixed solvent with CDCl$_3$ was used as a solvent. MALDI-TOF mass data were recorded on an AB SCIEX TOF/TOF 5800 system. ESI-TOF mass data were recorded on a Waters SYNAPT G2 HDMS system or an AB SCIEX TripleTOF 4600 system. Ultraviolet (UV)–Vis spectra were recorded on a JASCO V-670 spectrophotometer. Emission spectra were recorded on a JASCO FP-8600 fluorescence spectrophotometer. Absolute fluorescence quantum yields were determined with a Hamamatsu Photonics absolute PL quantum yield measurement system C9920-02. Solvents used for measurements were air-saturated. IR spectra were recorded on a JASCO FT/IR-480Plus spectrometer. Elemental analysis was performed on a Yanaco MT-6 analyzer with tin boats purchased from Elementar. We appreciate Mr Ikuo Iida of University of Tsukuba for the elemental analysis.

**Synthesis of hexapap H$_6$1**. A solution of **2** (48.2 mg, 0.156 mmol, 1.0 eq.) and $p$-TsOH·H$_2$O (6.0 mg, 0.03 mmol, 0.2 eq.) in a THF/H$_2$O = 3/1 mixed solvent (4 mL) was stirred for 10 h at 60 °C under Ar atmosphere. The reaction mixture was neutralized with sat. NaHCO$_3$ aq. (5.0 mL), and the precipitation was collected by filtration. The solid was washed with H$_2$O, CH$_3$CN, and MeOH, and dried in vacuo to give H$_6$1·6H$_2$O as a yellow solid (29.6 mg, 18.26 μmol, 72%).

mp: > 280 °C; $^1H$ NMR (600 MHz, CDCl$_3$/CD$_3$OD = 10/1 (v/v)): $\delta$ 9.12 (d, $J$ = 2.0 Hz, 6H), 8.91 (s, 6H), 8.32 (dd, $J$ = 8.1, 2.0 Hz, 6H), 8.14 (d, $J$ = 8.1 Hz, 6H), 7.56 (d, $J$ = 2.2 Hz, 6H), 7.46 (d, $J$ = 2.2 Hz, 6H), 1.44 (s, 54H); IR (Nujol): 3300 (m, OH), 1623 (m, C = N), 1578 (w), 1303 (m), 1260 (s), 1227 (m), 1202 (m), 1096 (w), 1021 (w), 955 (m), 861 (m), 838 (m), 738 (w), 636 (m) cm$^{-1}$; UV/Vis (CHCl$_3$/CH$_3$OH = 10/1 (v/v)): $\lambda_{max}$ 398 nm; MALDI TOF MS ($m/z$): [H$_6$1·Na$^+$] calcd. for C$_{96}$H$_{96}$N$_{12}$O$_6$Na, 1535.75; found, 1535.72; analysis (calcd., found for C$_{96}$H$_{108}$N$_{12}$O$_{12}$ (H$_6$1·6H$_2$O)): C (71.09, 71.39), H (6.71, 6.57), N (10.36, 10.40). It was difficult to obtain a good $^{13}C$ NMR spectrum due to low solubility of H$_6$1.

**Complexation of hexapap H$_6$1 and Zn(acac)$_2$**. H$_6$1 (1.00 mg, 0.66 μmol, 1.0 eq.) and Zn(acac)$_2$ (1.05 mg, 3.98 μmol, 6.0 eq.) in a CDCl$_3$/CD$_3$OD = 10/1 mixed solvent (500 μL) were mixed at room temperature. The complexation reaction was completed within 5 min and the formation of Zn-hexapap $[1Zn_6(acac)_6]$ was confirmed by $^1H$ NMR, UV–Vis absorption and emission measurements (Fig. 1f, h).

$^1H$ NMR (600 MHz, CDCl$_3$/CD$_3$OD = 10/1 (v/v)): $\delta$ 9.31 (s, 6H), 8.68 (s, 6H), 7.95 (dd, $J$ = 7.9, 1.5 Hz, 6H), 7.68 (dd, $J$ = 7.9, 1.5 Hz, 6H), 7.57 (d, $J$ = 2.2 Hz, 6H), 7.32 (d, $J$ = 2.2 Hz, 6H), 1.38 (s, 54H); UV/Vis (CHCl$_3$/CH$_3$OH = 10/1 (v/v)): 546 nm; emission (CHCl$_3$/CH$_3$OH = 10:1 (v/v)): 762 nm ($\lambda_{ex}$ = 546 nm); emission quantum yield: $\Phi_F$ = 0.017 ($\lambda_{ex}$ = 546 nm).

**Binding experiments of dicarboxylic acids with Zn-hexapap**. A representative procedure (pimelic acid H$_2$4e, Fig. 2e): H$_6$1 (1.00 mg, 0.66 μmol, 1.0 eq.) and Zn(acac)$_2$ (1.05 mg, 3.98 μmol, 6.0 eq.) in a CDCl$_3$/CD$_3$OD = 10/1 mixed solvent (500 μL) were mixed at room temperature.The formation of Zn-hexapap $[1Zn_6(acac)_6]$ was checked by $^1H$ NMR. To the solution was added pimelic acid H$_2$4e (2.0 μmol, 3.0 eq. for H$_6$1) in a CDCl$_3$/CD$_3$OD = 10/1 mixed solvent (3.0 μL). The reaction mixture was heated at 50 °C for 2 h. The resulting host–guest complexes were characterized by $^1H$ NMR, UV–Vis absorption and emission, and ESI-TOF-MS measurements.

**X-ray crystallographic analysis of $[1_2Zn_{12}4e_2(H_2O)_4Cl_8]$**. To the microtube charged with H$_6$1 (2.0 mg, 1.32 μmol, 1.0 eq.) and 1,1,2,2-tetrachloroethane/methanol = 10/1 (v/v) (500 μL) was added Zn(acac)$_2$ (2.09 mg, 7.96 μmol, 6.0 eq.) and pimelic acid H$_2$4e (0.64 mg, 3.96 μmol, 3.0 eq.). The reaction mixture was heated for 3 h at 50 °C. Single crystal of $[1_2Zn_{12}4e_2(H_2O)_4Cl_8]$ suitable for X-ray diffraction analysis was obtained by slow diffusion of acetone vapor into the solution.

The diffraction intensity data was measured at 100 K using MAR-CCD equipped at BL26B2 SPring-8[27]. The wavelength of incident X-ray was 0.8 Å. The collected diffraction images were processed by CrystalClear (Rigaku). The initial structure was solved using SIR92[28] and refined using SHELXL-2016[29]. The diffraction data up to 0.9 Å was used for the structure refinement.

Crystal data for $[1_2Zn_{12}4e_2(H_2O)_4Cl_8]\cdot1.5C_2H_2Cl_4$: C$_{209}$H$_{211}$Cl$_{14}$N$_{24}$O$_{24}$Zn$_{12}$, $Fw$ = 4723.99, purple prism, $0.10 \times 0.07 \times 0.05$ mm$^3$, tetragonal, space group $P4_1$ (No. 76), $a$ = 30.029(3) Å, $c$ = 40.154(5) Å, $V$ = 36209(9) Å$^3$, $Z$ = 4, $T$ = 100 K, $\lambda$ = 0.800 Å, $\theta_{max}$ = 26.387°, $R_1$ = 0.0981, $wR_2$ = 0.2946 (after SQUEEZE[30]), GOF = 1.038.

**Control of binding of a dicarboxylic acid by acid/base stimuli**. Experimental procedure (Fig. 4a, Supplementary Fig. 20): Hexapap H$_6$1 (1.96 mg, 1.29 μmol, 1.0 eq.) was weighed in an NMR tube. To the tube were added Zn(acac)$_2$ (2.13 mg, 8.08 μmol, 6 eq.) in a CDCl$_3$/CD$_3$OD = 10/1 mixed solvent (550 μL) and pimelic acid H$_2$4e (2.6 μmol, 2 eq. for H$_6$1) in a CDCl$_3$/CD$_3$OD = 10/1 mixed solvent (4.0 μL). The reaction mixture was heated at 50 °C for 3.5 h to produce $[1_2Zn_{12}4e_2X_n]$. To the solution was added CF$_3$SO$_3$H (4.0 μmol, 3 eq. for H$_6$1) in a CDCl$_3$/CD$_3$OD = 10/1 mixed solvent (6.0 μL). The conversion of the host–guest complexes from $[1_2Zn_{12}4e_2X_n]$ to $[1_2Zn_{12}4e_4X_n]$ was characterized by $^1H$ NMR and ESI-TOF-MS measurements. To the solution was added Me$_4$NOH·5H$_2$O (18.1 μmol, 14 eq. for H$_6$1) in CD$_3$OD (28 μL), which released the $4e^{2-}$ and produced guest-free Zn-hexapap $[1Zn_6X_n]$.

**X-ray crystallographic analysis of $[1_2Zn_{12}4e_4(H_2O)_4Cl_4]\cdot2C_2H_2Cl_4$**. To a 20 mL flask charged with H$_6$1 (25 mg, 16.5 μmol, 1.0 eq.) and CHCl$_3$/CH$_3$OH = 10/1 (v/v) (6 mL) was added Zn(acac)$_2$ (26.2 mg, 100 μmol, 6.1 eq.) and pimelic acid H$_2$4e (7.9 mg, 49 μmol, 3.0 eq.). The mixture was stirred at 55 °C for 2 days. The solution was filtered through a membrane filter, and to the filtrate was diffused the vapor of isopentane at 4 °C. The resultant precipitate was collected by filtration and dried to yield a purple solid (31.9 mg). The obtained complex was dissolved in 1,1,2,2-tetrachloroethane/methanol = 10/1 (v/v). Single crystal of $[1_2Zn_{12}4e_4(H_2O)_4Cl_4]\cdot2C_2H_2Cl_4$ suitable for X-ray diffraction analysis was obtained by slow diffusion of acetone vapor into the solution.

Single-crystal X-ray crystallographic measurements were performed using a Bruker APEX II ULTRA with MoKα radiation (graphite-monochromated, $\lambda$ = 0.71073 Å) at 120 K. The collected diffraction images were processed by Bruker APEX2. The initial structure was solved using SHELXS-97[31] and refined using SHELXL-2016[29]. The diffraction data up to 0.9 Å was used for the structure refinement.

Crystal data for $[1_2Zn_{12}4e_4(H_2O)_4Cl_4]\cdot2C_2H_2Cl_4$: C$_{224}$H$_{232}$Cl$_{12}$N$_{24}$O$_{32}$Zn$_{12}$, $Fw$ = 4982.42, purple block, $0.11 \times 0.11 \times 0.05$ mm$^3$, tetragonal, space group $I4_1/a$ (No. 88), $a$ = 29.8662(12) Å, $c$ = 40.741(2) Å, $V$ = 36341(3) Å$^3$, $Z$ = 4, $T$ = 120 K, $\lambda$(MoKα) = 0.71073 Å, $\theta_{max}$ = 23.282°, $R_1$ = 0.1014, $wR_2$ = 0.3590 (after SQUEEZE[30]), GOF = 1.032.

**Data availability**. CCDC 1507879 and 1507880 contain the data for this paper. The data can be obtained free of charge from The Cambridge Crystallographic Data Centre via www.ccdc.cam.ac.uk/getstructures. All the other data is available from the authors upon reasonable request.

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

## Acknowledgements

This research was supported by JSPS KAKENHI Grant Numbers JP15H00914, JP15H00723, JP26888003, JP17K14455, JP17H05351 (Coordination Asymmetry), TOBE MAKI Scholarship Foundation (16-JA-004), and Tokuyama Science Foundation. The synchrotron radiation experiments were performed at BL26B2 in SPring-8 with the approval of RIKEN (Proposal No. 20155498).

## Author contributions

T. Nakamura and T. Nabeshima conceived the project. T. Nakamura and Y.K. designed and analyzed the experiments. Y.K. carried out the experimental work. E.N. performed X-ray measurement and analysis. T. Nakamura and T. Nabeshima prepared the manuscript, and all the authors contributed to the writing of the paper.
