## [Peer Review File · Nature Communications]

Reviewers' comments:

Reviewer #1 (Remarks to the Author):

Crystallographic Review

It appears that the two structures have been worked on by different people, and there is considerable inconsistency between the structural refinements. The treatment of the two structures should be harmonized. For example, it should be noted that different target values have been used for restraints of equivalent bonds in the different structures - these should be consistent across the two structures.

As the structures currently stand, they should not be published, however they could be brought up to publication standard.

Structure 187a

Although this structure is essentially correct, there is a large number of issues with the structure refinement which need to be fixed before it is suitable for publication.

Firstly, ALL A- and B-level alerts should be explained through the use of appropriate `_vrf`'s.

It is unclear whether PLATON SQUEEZE has been applied. If it has not, it probably should be.

A large number of reflections have been individually omitted from the refinement. Only the few worst matching reflections should actually be omitted.

There must be an explanation of the disorder present, how it was modeled, and a further explanation of any other restraints used in the `_refine_special_details` section of the CIF. The authors should note that the ISOR restraint is intended for use only on isolated atoms. It should not be used for bonded atoms, in which case RIGU, DELU and SIMU are the appropriate restraints to use.

Atom O10 should be attached to C66 in the asymmetric unit, as this forms a complete molecule.

The tetrachloroethane molecule will probably be better refined as half-occupied.

Z has been reported as 16, when it should be 4 (Z' is 1/4, rather than 1).

The structure could probably benefit from a global rigid bond restraint, rather than the few selectively applied ones.

Structure shelx

This structure, as supplied, is chemically incorrect. The disorder of the diacid molecules has not been modeled in any sensible fashion. There is a number of incorrect bonds in the model, and the scattergun approach to addition of hydrogens is inappropriate. The authors should make use of appropriate PART

commands to correctly model this disorder, and should competitively refine the occupancies of the disorder components. Additionally, one disorder component of the diacid forms a covalent bond to an adjacent tetrachloroethane molecule. The tetrachloroethane molecule should be placed in an appropriate PART, and its occupancy refined complementary to that of the atom to which it is adjacent.

It should be noted that SQUEEZE has been correctly implemented in the refinement of this structure.

Again, all A- and B-level CIFCHECK alerts should be answered with _vrf's in the CIF. The following ALERTs, however, should be fixed, rather than explained:

SHFSU01_ALERT_2_A The absolute value of parameter shift to su ratio > 0.20

Absolute value of the parameter shift to su ratio given 1.625

Additional refinement cycles may be required.

PLAT080_ALERT_2_A Maximum Shift/Error 1.63 Why ?

- The structure refinement has not been finished. The structure must be refined to convergence.

PLAT412_ALERT_2_A Short Intra XH3 .. XHn H16E .. H238 .. 1.69 Ang.

- Antibumping restraints should be applied.

PLAT213_ALERT_2_B Atom C084 has ADP max/min Ratio 4.1 oblate

PLAT242_ALERT_2_B Low 'MainMol' Ueq as Compared to Neighbors of C140 Check

PLAT242_ALERT_2_B Low 'MainMol' Ueq as Compared to Neighbors of C270 Check

PLAT242_ALERT_2_B Low 'MainMol' Ueq as Compared to Neighbors of C291 Check

- Better thermal parameter restraints should be applied to the structure. A global application of ISOR is inappropriate.

PLAT780_ALERT_1_B Coordinates do not Form a Properly Connected Set Please Do !

- There is no reason not refine the structure as a single connected set.

Again, this structure needs an explanation of the disorder modeling, and application of restraints used in the _refine_special_details section of the CIF.

Reviewer #2 (Remarks to the Author):

In this manuscript, Nabeshima et al described the one-pot synthesis and binding properties of an imine macrocycle that contains multiple metal chelation sites inwardly. The structures of the imine macrocycle and its complexes with dicarboxylates were unambiguously characterized by ¹H NMR spectroscopy and single crystal X-ray diffraction, together with absorption and emission spectra.

The synthesis of the imine macrocycle is very efficient thanks to the judicious synthetic design. All binding sites are positioned inward and convergent to the center, thus affording high specificity and selectivity towards pimelic acid. The formation of a dimeric structure upon guest binding is very unique and nicely demonstrates the power and advantage of self-assembly or dynamic combinatorial library. Authors also showed that the binding mode could be reversibly switched by acid-base chemistry. I think this work is very creative and sufficiently novel for the publication in the Nature. Commun., and I recommend this manuscript published in this Journal essentially as it stands.

In the SI, p S27, line 12, there is a typo in the crystal data: ...(4e)4 ...should be (4e)2

Reviewer #3 (Remarks to the Author):

Novelty: Moderate

Impact: The work will be of great interest to the general chemistry community, but not to the general science community. We recommend publication to a high impact general chemistry journal.

A multinuclear complex capable of undergoing size specific guest-host interaction is described. 2D NMR experiments in conjunction with structural characterization via x-ray diffraction of crystals of the complexes show that the authors indeed made the complex and that it bound 2 substrates in a site specific manner. A unique recognition profile could be isolated using acidic conditions as stimuli. These unique profiles do not perform any chemical function nor do they contain novel binding motifs. The work is very interesting but ultimately not immediately of general utility.

General observations:

- ROESY experiments showed cross peaks that supported both the binding of the substrate and the binding of 2 macrocycles to make the "wavy-dimer"
- The crystal structure of the precursor and the guest bound complexes are consistent with the cross peaks observed by ROESY showing specific atoms brought near each other through space
- Characterization of synthesized materials is pristine.
- Manuscript contains a lot of run on sentences.

Suggested experiments and comments:

- Authors claim that the bimolecular recognition mode is incredibly stable since an excess of the ligand did not lead to changes observed by ¹H NMR. This would also be observed if rapid exchange was taking place. The best way to support this claim is to calculate a binding constant using concentration dependent studies. Absent that, the authors could add excess acetic acid or some other small mono acid and show that the dimer does not dissociate.
- The authors claim that binding is size specific in the regime of 4 to 5 carbon diacids noting an increase in fluorescence when those guests were introduced; this is not well supported as the NMR with the larger diacids showed multiple species forming likely because of oligomerization which would lead to self-quenching which is consistent with what the authors observed.

Responses to the reviewers' comments

Following the comments from the reviewers, we have revised our manuscript. *The reviewers' comments are cited in italics and in blue letters.* Word files of the manuscript and the supplementary information with “Track Changes” records have been separately uploaded.

To reviewer 1

First, we would like to thank the present reviewer for helpful suggestions in revising the manuscript. We carefully revised the manuscript and the cif files. According to the changes described below, Fig. 3 and Fig. 4b,c in the manuscript, Supplementary Figs. S14, S22, and S23 have been redrawn with the revised structures. Descriptions in the Methods section have been also revised.

Detailed replies and changes we have made are as follows:

Crystallographic Review

It appears that the two structures have been worked on by different people, and there is considerable inconsistency between the structural refinements. The treatment of the two structures should be harmonized. For example, it should be noted that different target values have been used for restraints of equivalent bonds in the different structures - these should be consistent across the two structures.

The two structures have been reanalyzed by the same crystallographer. Target values for restraints of equivalent bonds in the different structure are now consistent across the two structures. Molecular structures are similar for the two structures, but the space group is different. The number of independent atomic sites in crystallographic asymmetric unit of one structure ($P4_1$) is approximately four times larger than that of the other one ($I4_1/a$). So several different treatments were required for each analysis.

We would like to describe the scale of current crystallographic procedures. The molecular structures of the present study have approximately 300 atoms excluding hydrogens. Size and

scale of the molecule are fairly large for small-molecule crystallography. The size is comparable to that of small-size protein with less than 100 residues. In addition, there are large solvent accessible voids in the crystal structure similar to macromolecular crystallography.

As the structures currently stand, they should not be published, however they could be brought up to publication standard

.

Structure 187a

Although this structure is essentially correct, there is a large number of issues with the structure refinement which need to be fixed before it is suitable for publication.

Firstly, ALL A- and B-level alerts should be explained through the use of appropriate _vrf's.

We added appropriate _vrf's for all B-level alerts as suggested by the reviewer. All A-level alerts have been resolved after the reanalysis. After the checkCIF (ver. 26/Feb/2017), the responses in the vrf form that were intended for Alert level B also appear for the Alert level C tests under the same test number, but we suppose that this would be an error of the checkCIF program.

It is unclear whether PLATON SQUEEZE has been applied. If it has not, it probably should be.

We applied PLATON SQUEEZE in the revised refinement as suggested.

A large number of reflections have been individually omitted from the refinement. Only the few worst matching reflections should actually be omitted.

We have not omitted reflections in the revised refinement. There is a PLAT934 B-level alert in the checkcif report. All of these are low-order reflections. The data were measured using a diffractometer for small molecule crystallography. In addition, there are large void space in the crystal structure. These outliers mainly come from the difficulties of measurement and refinement.

There must be an explanation of the disorder present, how it was modeled, and a further explanation of any other restraints used in the _refine_special_details section of the CIF. The authors should note

that the ISOR restraint is intended for use only on isolated atoms. It should not be used for bonded atoms, in which case RIGU, DELU and SIMU are the appropriate restraints to use.

There is no disorder in the current structure. The explanations on applied restraints are given in the `_refine_special_details` section of the revised cif file. Detailed information on the refinement can be referred in the `_shelx_res_file` section. No ISOR restraint was applied in the current refinement.

Atom O10 should be attached to C66 in the asymmetric unit, as this forms a complete molecule.

This point was corrected as suggested.

The tetrachloroethane molecule will probably be better refined as half-occupied.

We did refinements using full- and half- occupancy models for the tetrachloroethane molecule. Judging from the refinements, we selected the half-occupied model as suggested reviewer.

Z has been reported as 16, when it should be 4 (Z' is 1/4, rather than 1).

We corrected Z value as suggested.

The structure could probably benefit from a global rigid bond restraint, rather than the few selectively applied ones.

Thank you for good suggestion. We cannot find more global rigid bond for this molecule.

Structure shelx

This structure, as supplied, is chemically incorrect. The disorder of the diacid molecules has not been modeled in any sensible fashion. There is a number of incorrect bonds in the model, and the scattergun approach to addition of hydrogens is inappropriate. The authors should make use of appropriate PART commands to correctly model this disorder, and should competitively refine the occupancies of the disorder components. Additionally, one disorder component of the diacid forms a covalent bond to an adjacent tetrachloroethane molecule. The tetrachloroethane molecule should be placed

in an appropriate PART, and its occupancy refined complementary to that of the atom to which it is adjacent.

Total number of individual atomic sites is approximately 300, which four times more than that of the other structure due to low crystallographic symmetry. The difficulty of the refinement is similar to that of small-sized macromolecular crystallography.

To determine chemically correct structures, we applied many kinds of constraints and restraints in the refinement. Many of the six-membered rings are constructed using AFIX 66 constraints. Almost all the hydrogen atoms were removed to achieve sufficient parameter shift to su ratio. Anti-bumping restraints were applied as suggested by the reviewer.

We carefully checked the converged structure, and confirmed that it did not contain any errors on structures and occupancies.

It should be noted that SQUEEZE has been correctly implemented in the refinement of this structure.

We also applied PLATON SQUEEZE in the revised refinement.

Again, all A- and B-level CIFCHECK alerts should be answered with _vrf's in the CIF. The following ALERTs, however, should be fixed, rather than explained:

All A-level alerts were fixed in the revised refinement. We also answered all B-level alerts as suggested by the reviewer. After the checkCIF (ver. 26/Feb/2017), the responses in the vrf form that were intended for Alert level B also appear for the Alert level C tests under the same test number, but we suppose that this would be an error of the checkCIF program.

SHFSU01_ALERT_2_A The absolute value of parameter shift to su ratio > 0.20

Absolute value of the parameter shift to su ratio given 1.625

Additional refinement cycles may be required.

PLAT080_ALERT_2_A Maximum Shift/Error 1.63 Why ?

- The structure refinement has not been finished. The structure must be refined to convergence.

PLAT412_ALERT_2_A Short Intra XH3 .. XHn H16E .. H238 .. 1.69 Ang.

- Antibumping restraints should be applied.

These are no A-level alerts in the revised structure.

PLAT213_ALERT_2_B Atom C084 has ADP max/min Ratio 4.1 oblate
PLAT242_ALERT_2_B Low 'MainMol' Ueq as Compared to Neighbors of C140 Check
PLAT242_ALERT_2_B Low 'MainMol' Ueq as Compared to Neighbors of C270 Check
PLAT242_ALERT_2_B Low 'MainMol' Ueq as Compared to Neighbors of C291 Check
- Better thermal parameter restraints should be applied to the structure. A global application of ISOR is inappropriate.

We still use global application of ISOR and SIMU restraints like macromolecular crystallography. Without these restraints, it is very difficult to determine a reliable and chemically correct structure due to low data quality and many parameters. To determine a chemically possible solution, we also applied XNPD restraints.

PLAT780_ALERT_1_B Coordinates do not Form a Properly Connected Set Please Do !
- There is no reason not refine the structure as a single connected set.

This problem was solved in the present structure.

Again, this structure needs an explanation of the disorder modeling, and application of restraints used in the `_refine_special_details` section of the CIF.

The disorder part of current structure comes from two different molecules. There are two combinations. One is (A) C172-C155-C260-C259-C257-C109-C229 and (B) C230-C116-C161-C325-C324-C091-C226. Another is (C) C230-C116-C161-C261-C247-C155-C172 and (D) C226-C091-C323-C321-C322-C109-C229. Thermal parameters of (A) and (C) are almost identical. (B) and (D) are also almost identical. So, we fix occupancies of disordered atoms to 0.5.

The explanations on applied restraints are given in the `_refine_special_details` section of the revised .cif file. Detailed information on the refinement can be referred in the `_shelx_res_file` section.

To Reviewer 2

In this manuscript, Nabeshima et al described the one-pot synthesis and binding properties of an imine macrocycle that contains multiple metal chelation sites inwardly. The structures of the imine macrocycle and its complexes with dicarboxylates were unambiguously characterized by ¹H NMR spectroscopy and single crystal X-ray diffraction, together with absorption and emission spectra. The synthesis of the imine macrocycle is very efficient thanks to the judicious synthetic design. All binding sites are positioned inward and convergent to the center, thus affording high specificity and selectivity towards pimelic acid. The formation of a dimeric structure upon guest binding is very unique and nicely demonstrates the power and advantage of self-assembly or dynamic combinatorial library. Authors also showed that the binding mode could be reversibly switched by acid-base chemistry. I think this work is very creative and sufficiently novel for the publication in the Nature. Commun., and I recommend this manuscript published in this Journal essentially as it stands.

Thank you very much for your kind comments on our manuscript. We appreciate your recommendation for publication of our work. We believe this paper contributes to the design strategy of functional supramolecular complexes, as well as the development of selective molecular sensors or allosteric multinuclear metal catalysts.

In the SI, p S27, line 12, there is a typo in the crystal data: ...(4e)4 ...should be (4e)2

Thank you for your comments. We corrected the corresponding part accordingly.

To Reviewer 3

Novelty: Moderate

Impact: The work will be of great interest to the general chemistry community, but not to the general science community. We recommend publication to a high impact general chemistry journal.

A multinuclear complex capable of undergoing size specific guest-host interaction is described. 2D NMR experiments in conjunction with structural characterization via x-ray diffraction of crystals of the complexes show that the authors indeed made the complex and that it bound 2 substrates in a site specific manner. A unique recognition profile could be isolated using acidic conditions as stimuli. These unique profiles do not perform any chemical function nor do they contain novel binding motifs. The work is very interesting but ultimately not immediately of general utility.

Thank you very much for your great interest in our paper. We appreciate your valuable comment. As we stated in the introduction, most biological and synthetic receptors employ a combination of weak intermolecular interactions such as hydrogen bonds, π - π interactions, and van der Waals interactions. Meanwhile, multiple coordination bonds are employed for molecular binding in this supramolecular system. This design strategy provides a fresh viewpoint not only to general chemists, but also to specialists with various academic disciplines including physics, biology, and medicines. Furthermore, although the current paper does not include immediate applications, the macrocycle capturing small molecules utilizing multiple coordination bonds leads to the next-generation allosteric metal catalysts and/or specific molecular sensors. Thus, we believe that this paper is suitable for publication in *Nature Communications*.

General observations:

- ROESY experiments showed cross peaks that supported both the binding of the substrate and the binding of 2 macrocycles to make the “wavy-dimer”*
- The crystal structure of the precursor and the guest bound complexes are consistent with the cross peaks observed by ROESY showing specific atoms brought near each other through space*
- Characterization of synthesized materials is pristine.*

Thank you very much for supporting our experimental results and analyses.

- *Manuscript contains a lot of run on sentences.*

According to your pointers, the following sentences have been revised.

- **Page 2, line 2,**

(Original) vital role in nature, and it serves as a basis

(Revised) vital role in nature. It serves as a basis

- **Page 2, lines 21–24,**

(Original) To utilize multiple coordination bonds in the molecular recognition events, labile coordination sites on the metal centers, to which external small molecules bind in place of exchangeable ligands, should be spatially arranged in the binding pocket.

(Revised) To utilize multiple coordination bonds in the molecular recognition events, labile coordination sites on the metal centers **should be spatially arranged in the binding pocket. External small molecules bind to multiple metals via coordination bonds in place of the exchangeable ligands.**

- **Page 3, line 4,**

(Original) by pap^- remain and are available

(Revised) by pap^- are available

- **Page 3, line 4,**

(Original) binding, while the chelation of the metal

(Revised) binding. **Meanwhile,** the **tridentate** chelation of the metal

- **Page 5, line 5,**

(Original) employed, and its derivative

(Revised) employed. **Its** derivative

- **Page 6, lines 2–3,**

(Original)), which indicated

(Revised)). **These results** indicated

- **Page 8, line 25,**

(Original) Zn atoms, one of which

(Revised) Zn atoms. **One of the Zn atoms**

· **Page 9, line 2,**

(Original) a water, thus free

(Revised) a water. Thus, Zn (type 3) was free

· **Page 9, line 11,**

(Original) $4e^{2-}$, and the

(Revised) $4e^{2-}$. The

· **Page 12, line 9,**

(Original) them, which is

(Revised) them. This is

· **Page 14, line 5,**

(Original) and resulted in the unique saddle-saddle shaped

(Revised) to form the unique saddle-shaped

Suggested experiments and comments:

· *Authors claim that the bimolecular recognition mode is incredibly stable since an excess of the ligand did not lead to changes observed by 1H NMR. This would also be observed if rapid exchange was taking place. The best way to support this claim is to calculate a binding constant using concentration dependent studies. Absent that, the authors could add excess acetic acid or some other small mono acid and show that the dimer does not dissociate.*

We appreciate your valuable suggestions. The host-guest complex of the bimolecular recognition mode $[1_2Zn_{12}4e_2X_n]$ has a C_2 point-group symmetry (6 different Zn-pap units), because the top and bottom macrocycles are in a different environment as the result of the binding of $4e^{2-}$. Meanwhile, the guest-free $[1_2Zn_{12}X_n]$ dimeric framework and the tetramolecular recognition mode $[1_2Zn_{12}4e_4X_n]$ have a S_4 point-group symmetry (3 different Zn-pap units). As seen from the Supplementary Fig. 17, the 1H NMR spectrum showed that the host-guest complex has a C_2 symmetry, thus it was confirmed that the Zn-complex existed as the bimolecular recognition mode $[1_2Zn_{12}4e_2X_n]$, and the two $4e^{2-}$ molecules were fixed in the cavity on the NMR timescale.

After careful reinvestigations of the 1H NMR spectra, however, we noticed that the possibility of the binding of *additional* $4e^{2-}$ onto the bimolecular recognition mode $[1_2Zn_{12}4e_2X_n]$ with rapid

exchange compared to the NMR timescale cannot be excluded. In other words, X, a labile ligand exchanging faster than the NMR timescale, can be an additional $4e^{2-}$ if excess amount of H_24e were added.

Based on the discussion above, the following descriptions have been revised.

• **Page 11, lines 3–6,**

(Original) the addition of more than 2 molar amounts of the pimelic acids H_24e against the wavy-stacked dimer $[1_2Zn_{12}X_n]$ did *not* result in further binding, but the host-guest complex with two $4e^{2-}$ s, $[1_2Zn_{12}4e_2X_n]$, stably existed (Supplementary Fig. 17).

(Revised) the addition of more than 2 molar amounts of the pimelic acids H_24e against the wavy-stacked dimer $[1_2Zn_{12}X_n]$ did **not change a binding mode**, but the host-guest complex **stably existed in a bimolecular recognition mode** $[1_2Zn_{12}4e_2X_n]$ (see Supplementary Fig. 17).

• **Page 11, lines 7–8,**

(Original) lead to the complex with four $4e^{2-}$ s, $[1_2Zn_{12}4e_4X_n]$ (Fig. 4a).

(Revised) led to a **tetramolecular recognition mode** $[1_2Zn_{12}4e_4X_n]$ (Fig. 4a, see Supplementary Fig. 18).

• **Page 13,**

Fig. 4a has been revised so as not to include free H_24e together with bimolecular recognition mode $[1_2Zn_{12}4e_2X_n]$. Furthermore, the molecular formula in Fig. 4 has been changed from $[1_2Zn_{12}4e_2(ROH)_{12-m}(RO)_m]^{n+}$ and $[1_2Zn_{12}4e_4(ROH)_{8-m}(RO)_m]^{n+}$ to $[1_2Zn_{12}4e_2X_n]$ and $[1_2Zn_{12}4e_4X_n]$, respectively, since the possibility of the binding of additional $4e^{2-}$ as a rapidly exchanging ligand cannot be ruled out. The schematic representations of dimers $[1_2Zn_{12}4e_2X_n]$ and $[1_2Zn_{12}4e_4X_n]$ have been also modified in the following two points: 1) The structure has been slightly tilted to emphasize its dimeric structure; 2) ROH or RO^- ligands, depicted as green circular cylinders coordinating to the Zn atoms of the dimer, have been omitted.

(Original)

(Revised)

• Page S17, in the caption of Supplementary Fig. 17

(Original) Addition of excess amount of H₂4e did not result in further binding to [1₂Zn₁₂4e₂X_n].

(Revised) (deleted)

· **Page S18, in the caption of Supplementary Fig. 18**

(Original) Dimer of Zn-**hexapap** with two pimelates,

(Revised) Dimer of Zn-**hexapap** in bimolecular recognition mode,

(Original) Dimer of Zn-**hexapap** with four pimelates,

(Revised) Dimer of Zn-**hexapap** in tetramolecular recognition mode,

· *The authors claim that binding is size specific in the regime of 4 to 5 carbon diacids noting an increase in fluorescence when those guests were introduced; this is not well supported as the NMR with the larger diacids showed multiple species forming likely because of oligomerization which would lead to self-quenching which is consistent with what the authors observed.*

Thank you very much for your pointers. We did not intend to claim that the binding of dicarboxylates is size-specific, but intended to claim that the “wavy-dimer” structure is formed as a single species only when a specific size of carboxylates binds to the Zn-**hexapap**. We agree that the dicarboxylic acids other than adipic acid H₂**4d** and pimelic acid H₂**4e** also bind to the Zn-**hexapap**; they do not lead to the selective formation of “wavy-dimer”, but to the formation of multiple species (¹H NMR, Figs. 2a–2i). In the original manuscript, “recognition of specific dicarboxylic acids” was used to describe that only H₂**4d** and H₂**4e** were bound to Zn-**hexapap** in the bimolecular recognition mode [1₂Zn₁₂**4X_n**] as describe in Fig. 3. However, this phrase might be interpreted as that only the specific dicarboxylic acids can interact with Zn-**hexapap**. To avoid this confusion, we would like to revise the description in relation to “recognition of specific dicarboxylic acids” as below.

· **Page 1, lines 16–18,**

(Original) The metallomacrocycle recognized a specific length of dicarboxylic acids via multipoint coordination bonding accompanied by the formation of a unique wavy-stacked structure.

(Revised) The metallomacrocycle forms a unique wavy-stacked structure upon binding a suitable length of dicarboxylic acids via multipoint coordination bonding.

· **Page 2, line 8,**

(Original) To achieve selective binding,

(Revised) To achieve sophisticated recognition events,

· **Page 3, line 8,**

(Original) a sufficiently specific artificial host

(Revised) a **sophisticated** artificial host

· **Page 3, lines 11–13,**

(Original) the dicarboxylic acids with specific chain lengths were recognized via multiple coordination bonds between the carboxylate groups and Zn, and induced the formation of a uniquely-shaped wavy-stacked dimer of the Zn-**hexapap**

(Revised) the dicarboxylic acids with **suitable** chain lengths **induced the formation of a uniquely-shaped wavy-stacked dimer of the Zn-hexapap via multiple coordination bonds between the carboxylate groups and Zn**

· **Page 6, lines 11–14,**

(Original) The specific recognition was also supported by the change in fluorescence, where the samples in which H₂**4d** or H₂**4e** was mixed with [1Zn₆(acac)₆] showed a stronger red emission than the other dicarboxylic acids (Fig. 21).

(Revised) **The formation of a certain host-guest complex** was also supported by the change in **emission**, where the samples in which H₂**4d** or H₂**4e** was mixed with [1Zn₆(acac)₆] showed a stronger red emission than the other dicarboxylic acids (Fig. 21).

(We replaced the “fluorescence” with “emission” because we have not yet measured the emission lifetime.)

· **Page 7, lines 2–3,**

(Original) Specific molecular recognition of dicarboxylic acids accompanied by the formation of the wavy-stacked dimer.

(Revised) **Binding** of dicarboxylic acids **by Zn-hexapap and** the formation of the wavy-stacked dimer.

· **Page 14, line 4,**

(Original) Zn-**hexapap** specifically recognized.

(Revised) Zn-**hexapap** recognized

Reviewers' comments:

Reviewer #1 (Remarks to the Author):

As they stand, these structures should not be published. Although the structural quality has improved somewhat, there are still significant structural issues (especially with the P41 structure). Structure 1 (P41 structure):

There is a number of problems with this structure still.

The most significant problems, which must be addressed before this structure can be published: Hydrogens are missing from the large proportion of the structure. These should be included. If the authors are having problems with rotating hydrogen atoms, they should firstly refine with fixed methyl groups (AFIX 33 rather than AFIX 137), and if necessary refine the water molecules without hydrogen atoms – however if possible hydrogen bond interactions are considered, it may be possible to apply appropriate restraints to locate these as well.

More seriously, the disorder of the central bridging ligands has still not been modelled. This is so serious such that when a casual observer looks at the crystal structure, they cannot discern the nature of these bridging ligands. These bridging ligands (or large macrocycle as it is currently modelled), also form two bonds to the chlorine atom of an adjacent solvent molecule. If the authors are unsure how to correctly model disorder they should consult another crystallographer. The authors have use a wide array of thermal parameter restraints, and attempted a very fine-grained approach. The collection of SIMU, RIGU and ISOR restraints can all be replaced with simply

RIGU

SIMU 0.01 0.02 2

which will appropriately restrain the entire structure.

Other, the minor problems:

_cell_angle_alpha 90.000(5)

_cell_angle_beta 90.000(5)

_cell_angle_gamma 90.000(5)

Cell angles are typically quoted without uncertainties when the angle has a defined value (which it does in this case).

_computing_data_collection ?

_computing_cell_refinement ?

_computing_data_reduction ?

_computing_structure_solution ?

_atom_sites_solution_primary ?

_atom_sites_solution_secondary ?

These data items above should be completed.

_computing_molecular_graphics ?

_computing_publication_material ?

These data items should be completed if appropriate.

_vrf_THETM01_shelx

;

PROBLEM: The value of $\sin(\theta_{\max})/\text{wavelength}$ is less than 0.575

Calculated $\sin(\theta_{\max})/\text{wavelength} = 0.5555$

RESPONSE: This is due to the large molecular size and the solvents.

We believe that the quality of crystal is good from a macromolecular point of view but slightly insufficient for the criteria of small molecule crystallography.

;

The authors have not correctly interpreted what this ALERT is asking them about. Although the response could be considered vaguely correct, it does not address that the ALERT is asking about the data limit applied during the structure refinement.

_vrf_PLAT242_shelx

;

PROBLEM: Low 'MainMol' Ueq as Compared to Neighbors of C210 Check

RESPONSE: C210 is located between three methyl groups and a six-membered ring. Three methyl groups are located at the surface of the molecule and they are affected by strong thermal motion. The thermal motion of the six-membered ring is much smaller than those of the methyl groups.

;

Examining the three methyl groups, it is obvious that they are exhibiting strong rotational thermal motion. The central atom to which they are attached would not be expected to exhibit this thermal motion.

Structure 2 (I41/a)

This structure appears to be largely correct, although there are some improvements that should be made. Atom O10 should be connected to C66 in the asymmetric unit. The two parts of the outer ligand should be joined together in the asymmetric unit. The rotational disorder of the t-butyl groups could also be modelled.

There are additional minor errors:

_computing_data_collection ?

_computing_cell_refinement ?

_computing_data_reduction ?

_computing_structure_solution ?

_atom_sites_solution_primary ?

_atom_sites_solution_secondary ?

These data items above should be completed.

_computing_molecular_graphics ?

_computing_publication_material ?

These data items should be completed if appropriate.

_vrf_THETM01_shelx

;

PROBLEM: The value of $\sin(\theta_{\max})/\lambda$ is less than 0.575

Calculated $\sin(\theta_{\max})/\lambda = 0.5561$

RESPONSE: This is due to the large molecular size and the solvents.

We believe that the quality of crystal is good from a macromolecular point of view but slightly insufficient for the criteria of small molecule crystallography.

;

The authors have not correctly interpreted what this ALERT is asking them about. Although the response could be considered vaguely correct, it does not address that the ALERT is asking about the data limit applied during the structure refinement.

_vrf_PLAT934_shelx

;

PROBLEM: Number of $(I_{\text{obs}} - I_{\text{calc}})/\sigma W > 10$ Outliers 7 Check

RESPONSE: This is mainly due to the solvents. All of them are low-order ($d > 5.0\text{\AA}$) reflections. Their intensities highly are highly affected by the solvents.

;

These reflections should be OMITted from the refinement.

Reviewer #3 (Remarks to the Author):

Molecular recognition by multiple coordination inside wavy stacked macrocycles

Revised Manuscript: Ready for publication in Nature Comm.

From original review:

- Manuscript does not contain but does contain a lot of run on sentences.

Manuscript has been revised and is much more clear and concise as well as grammatically correct.

Suggested Experiments and comments:

- Authors claim that the bimolecular recognition mode is incredibly stable since an excess of the ligand didn't lead to changes by ^1H NMR. This would also be observed if rapid exchange was taking place. The best way to support this claim is to calculate a binding constant using concentration dependent studies. Absent that, the authors could add excess acetic acid or some other small mono acid and show that the dimer does not dissociate.

The combination of the supporting figure 17&18 along with the titrations of pimelic acid is sufficient to show that the binding mode is changed which speaks to the stability of the wavy stacked dimer. The revised Figure 4 is more concise.

- The authors claim that binding is size specific in the regime of 4 to 5 carbon diacids noting an increase in fluorescence when those guests were introduced; this is not well supported as the NMR with the larger diacids showed multiple species forming likely because of oligomerization which would lead to self-quenching which is consistent with what the authors observed.

Authors revised this claim to clarify that 4 and 5 carbon diacids led to sophisticated and clearly defined species as opposed to the possible mixtures that other linker lengths made. The authors also clarified that the decrease in fluorescence may have been due to self-quenching, and that they did not do lifetime studies to route this out. This revision is acceptable as the 4 to 5 carbon diacid bound structures were well characterized and that is more interesting than whether the host is specific to substrate length.

Conclusion:

Overall the revised manuscript adequately addressed our concerns and is satisfactory for publication.

Responses to the reviewers' comments

Following the comments from the reviewers, we have revised our manuscript. *The reviewers' comments are cited in italics and in blue letters.* Word files of the manuscript and the supplementary information with “Track Changes” records (from the 1st revision) have been separately uploaded.

To reviewer 1

As they stand, these structures should not be published. Although the structural quality has improved somewhat, there are still significant structural issues (especially with the P41 structure).

Structure 1 (P41 structure):

There is a number of problems with this structure still.

The most significant problems, which must be addressed before this structure can be published:

Hydrogens are missing from the large proportion of the structure. These should be included. If the authors are having problems with rotating hydrogen atoms, they should firstly refine with fixed methyl groups (AFIX 33 rather than AFIX 137), and if necessary refine the water molecules without hydrogen atoms – however if possible hydrogen bond interactions are considered, it may be possible to apply appropriate restraints to locate these as well.

We appreciate your careful review. Missing hydrogens have been added in the structure according to your advice. The hydrogens at the methylene groups belonging to the “branched part” of the central dicarboxylates were not modelled due to the limitation of the SHELX program.

More seriously, the disorder of the central bridging ligands has still not been modelled. This is so serious such that when a casual observer looks at the crystal structure, they cannot discern the nature of these bridging ligands. These bridging ligands (or large macrocycle as it is currently modelled), also form two bonds to the chlorine atom of an adjacent solvent molecule. If the authors are unsure how to correctly model disorder they should consult another crystallographer.

We modeled the central bridging ligands using PART instruction of shelxl. We also treated the disorder of solvent which located close to the central ligands. No bumping between molecules is observed in the present structure.

The authors have use a wide array of thermal parameter restraints, and attempted a very fine-grained approach. The collection of SIMU, RIGU and ISOR restraints can all be replaced with simply RIGU

SIMU 0.01 0.02 2

which will appropriately restrain the entire structure.

We tried to replace SIMU RIGU and ISOR restraints. The trials were not successful. So, we still use many restraints.

Other, the minor problems:

_cell_angle_alpha 90.000(5)

_cell_angle_beta 90.000(5)

_cell_angle_gamma 90.000(5)

Cell angles are typically quoted without uncertainties when the angle has a defined value (which it does in this case).

The angles were change to 90.0 without uncertainties.

_computing_data_collection ?

_computing_cell_refinement ?

_computing_data_reduction ?

_computing_structure_solution ?

_atom_sites_solution_primary ?

_atom_sites_solution_secondary ?

These data items above should be completed.

_computing_molecular_graphics ?

_computing_publication_material ?

These data items should be completed if appropriate.

These data items were filled in.

_vrf_THETM01_shelx

;

PROBLEM: The value of $\sin(\theta_{\max})/\lambda$ is less than 0.575

Calculated $\sin(\theta_{\max})/\lambda = 0.5555$

RESPONSE: This is due to the large molecular size and the solvents.

We believe that the quality of crystal is good from a macromolecular point of view but slightly insufficient for the criteria of small molecule crystallography.

The authors have not correctly interpreted what this ALERT is asking them about. Although the response could be considered vaguely correct, it does not address that the ALERT is asking about the data limit applied during the structure refinement.

We added the following two sentences.

There are no available reflections for analysis in $d < 0.9 \text{ \AA}$ region.

The signal to noise ratio is less than 1.5 in $0.85 < d < 0.9 \text{ \AA}$ in the data.

_vrf_PLAT242_shelx

;

PROBLEM: Low 'MainMol' Ueq as Compared to Neighbors of C210 Check

RESPONSE: C210 is located between three methyl groups and a six-membered ring. Three methyl groups are located at the surface of the molecule and they are affected by strong thermal motion.

The thermal motion of the six-membered ring is much smaller than those of the methyl groups.

;

Examining the three methyl groups, it is obvious that they are exhibiting strong rotational thermal motion. The central atom to which they are attached would not be expected to exhibit this thermal motion.

We fully agree with your comment. We revised the response as follows;

These atoms were located between three methyl groups of a tBu group and a six-membered ring.

Three methyl groups are exhibiting strong rotational thermal motion. The thermal motion of the central carbon atom is much smaller than those of the methyl groups.

Structure 2 (I41/a)

This structure appears to be largely correct, although there are some improvements that should be

made. Atom O10 should be connected to C66 in the asymmetric unit. The two parts of the outer ligand should be joined together in the asymmetric unit. The rotational disorder of the t-butyl groups could also be modelled.

We changed an atomic coordinate of O10 to be connected to C66. The two parts of the outer ligand has been joined together. We also modelled the rotational disorder of two of the t-butyl groups.

There are additional minor errors:

_computing_data_collection ?

_computing_cell_refinement ?

_computing_data_reduction ?

_computing_structure_solution ?

_atom_sites_solution_primary ?

_atom_sites_solution_secondary ?

These data items above should be completed.

_computing_molecular_graphics ?

_computing_publication_material ?

These data items should be completed if appropriate

These data items were filled in.

_vrf_THETM01_shelx

;

PROBLEM: The value of $\sin(\theta_{max})/\text{wavelength}$ is less than 0.575

Calculated $\sin(\theta_{max})/\text{wavelength} = 0.5561$

RESPONSE: This is due to the large molecular size and the solvents.

We believe that the quality of crystal is good from a macromolecular point of view but slightly insufficient for the criteria of small molecule crystallography.

;

The authors have not correctly interpreted what this ALERT is asking them about. Although the response could be considered vaguely correct, it does not address that the ALERT is asking about the data limit applied during the structure refinement.

We added the following sentence.

There are no available reflections for structure analysis in $d < 0.9$ Å.

_vrf_PLAT934_shelx

;

PROBLEM: Number of (Iobs-Icalc)/SigmaW > 10 Outliers 7 Check

RESPONSE: This is mainly due to the solvents. All of them are low-order ($d > 5.0$ Å) reflections. Their intensities highly are highly affected by the solvents.

;

These reflections should be OMITted from the refinement.

We omitted these reflections from the refinement.

According to the changes described above, Fig. 3 and Fig. 4b,c in the manuscript, Supplementary Figs. S14, S22, and S23 have been redrawn with the revised structures. Descriptions in the Methods section have been also revised.

To Reviewer 3

Molecular recognition by multiple coordination inside wavy stacked macrocycles

Revised Manuscript: Ready for publication in Nature Comm.

Thank you very much. We appreciate your recommendation for publication of our work in *Nature Communications*.

From original review:

- *Manuscript does not contain but does contain a lot of run on sentences.*

Manuscript has been revised and is much more clear and concise as well as grammatically correct.

We appreciate your advice to improve the manuscript.

Suggested Experiments and comments:

- *Authors claim that the bimolecular recognition mode is incredibly stable since an excess of the ligand didn't lead to changes by ¹H NMR. This would also be observed if rapid exchange was taking place. The best way to support this claim is to calculate a binding constant using concentration dependent studies. Absent that, the authors could add excess acetic acid or some other small mono acid and show that the dimer does not dissociate.*

The combination of the supporting figure 17&18 along with the titrations of pimelic acid is sufficient to show that the binding mode is changed which speaks to the stability of the wavy stacked dimer.

The revised Figure 4 is more concise.

Thank you very much for analyzing the results of experiments in detail. We really appreciate your support.

- *The authors claim that binding is size specific in the regime of 4 to 5 carbon diacids noting an increase in fluorescence when those guests were introduced; this is not well supported as the NMR with the larger diacids showed multiple species forming likely because of oligomerization which would lead to self-quenching which is consistent with what the authors observed.*

Authors revised this claim to clarify that 4 and 5 carbon diacids led to sophisticated and clearly defined species as opposed to the possible mixtures that other linker lengths made. The authors also clarified that the decrease in fluorescence may have been due to self-quenching, and that they did not do lifetime studies to route this out. This revision is acceptable as the 4 to 5 carbon diacid bound structures were well characterized and that is more interesting than whether the host is specific to

substrate length.

Conclusion:

Overall the revised manuscript adequately addressed our concerns and is satisfactory for publication.

Thanks to your good advice, the manuscript has been revised to be more accurate and more concise. We really appreciate your thorough review. We hope our study contributes to the wide range of scientific communities.

Reviewers' comments:

Reviewer #1 (Remarks to the Author):

Additional corrections are still needed for these structures.

Quote "The hydrogens at the methylene groups belonging to the "branched part" of the central dicarboxylates were not modelled due to the limitation of the SHELX program." - this is not true. These hydrogens can be modelled with SHELXL.

Quote "We tried to replace SIMU RIGU and ISOR restraints. The trials were not successful. So, we still use many restraints." - following the review is a copy of the converged structure where global RIGU and SIMU restraints have been applied, and the hydrogen atoms on the methylene atoms have been correctly modelled. Note that this refinement method removes the five B-level alerts relating to low Ueq values, and results in more sensible anisotropic thermal parameters over the whole model. It also removes the need for the XNPD restraint, and removes the need to DAMP the structure refinement.

Although the _vrf's now have more sensible comments, they are now no longer syntactically correct. For example, "_vrf_PLAT242_shelx" has become "_vrf_PLAT242". Omitting the _shelx identifier means the _vrf is no longer associated with the _shelx data block, and the _vrf no longer appears in the checkCIF report. This should be corrected for both structures.

Minor correction -

_diffn_radiation_monochromator 'Synchrotron' - should read 'Si double crystal monochromator'

Regarding the second structure:

Quote "We changed an atomic coordinate of O10 to be connected to C66. The two parts of the outer ligand has been joined together. We also modelled the rotational disorder of two of the t-butyl groups." The reviewer must apologize, there is also rotational disorder of the third t-butyl group, which needs to be modelled.

For this structure, again, refining with global RIGU and SIMU restraints (the SIMU slightly tightened to "SIMU 0.01 0.02 2") results in overall better quality thermal parameters, and removes the need entirely for any DAMP and XNPD restraints, and EADP constraints. An updated SHELX file has been appended to the report, with thermal parameter constraints and disorder modelled.

Additional points from the manuscript:

Page 19, line 16: "The diffraction intensity data up to 1.00 Å was measured..." - in the structure refinement the applied data limit was 0.9 Å. It is also implied in the CIF that data was collected to a higher angle, however was cut down during structure refinement do to poor quality of diffraction at higher angles. The manuscript should be updated to reflect this.

Page 19, line 18: "The structure was determined and refined using SHELXL-2014" - SHELXL-2016 is the version used to refine the structure, and the reference should be updated to the 2015 paper (<http://dx.doi.org/10.1107/S2053229614024218>). SHELXL is not a structure solution program. The program used to solve the structure (SIR92) should be included and appropriately referenced.

Page 20, line 23: "Obtained data were processed using a Bruker APEX2 and refined using SHELXL-2014." - The SHELXL version should be correct, and the reference updated. The program used to solve the structure (SHELXS) should be mentioned and appropriately reference (in this case the 2008 paper is the correct reference - <https://doi.org/10.1107/S0108767307043930>).

Updated SHELX file:

TITL SIR92 run in space group P 41

CELL 0.8 30.029 30.029 40.154 90 90 90

ZERR 4 0.003 0.003 0.005 0 0 0

LATT -1

SYMM -X,-Y,0.5+Z

SYMM -Y,+X,0.25+Z

SYMM +Y,-X,0.75+Z

SFAC C N O Zn Cl H

DISP C 0 0 9.49

DISP H 0 0 1

DISP N 0 0 19.35

DISP O 0 0 35.46

UNIT 836 96 96 48 56 780

DFIX 1.27 0.05 O037 C314

DFIX 1.51 0.01 C255 C254 C115 C246

DFIX 1.77 0.01 C115 Cl24 C115 Cl21 C246 Cl22 C246 Cl19

DFIX 1.77 0.01 C255 Cl25 C255 Cl26 C254 Cl27 C254 Cl28

DFIX 1.54 0.05 C221 C164 C221 C249 C221 C235

DFIX 2.5 0.05 C249 C164 C235 C249 C164 C235

DFIX 1.54 0.05 C301 C113 C301 C305 C301 C202

DFIX 2.5 0.05 C202 C113 C113 C305 C305 C202

DFIX 1.67 0.05 C301 C287

DFIX 1.54 0.05 C094 C220 C094 C222 C094 C157

DFIX 2.5 0.05 C157 C220 C220 C222 C222 C157

DFIX 1.54 0.05 C232 C215 C232 C234 C232 C236

DFIX 2.5 0.05 C234 C215 C236 C234 C215 C236

DFIX 1.54 0.05 C250 C248 C250 C253 C250 C238

DFIX 2.5 0.05 C253 C248 C238 C253 C248 C238

DFIX 1.54 0.05 C210 C244 C210 C252 C210 C237

DFIX 2.5 0.05 C252 C244 C237 C252 C244 C237

DFIX 1.54 0.05 C140 C233 C140 C217 C140 C243

DFIX 2.5 0.05 C217 C233 C243 C217 C233 C243

DFIX 1.54 0.05 C251 C242 C251 C160 C251 C199

DFIX 2.5 0.05 C160 C242 C199 C160 C242 C199

DFIX 1.67 0.05 C251 C240

DFIX 1.54 0.05 C218 C204 C218 C241 C218 C245

DFIX 2.5 0.05 C241 C204 C245 C241 C204 C245

DFIX 1.54 0.05 C216 C219 C216 C224 C216 C228

DFIX 2.5 0.05 C224 C219 C228 C224 C219 C228

DFIX 1.67 0.05 C216 C167

DFIX 1.54 0.05 C302 C304 C302 C306 C302 C163

DFIX 2.5 C306 C304 C163 C306 C304 C163

DFIX 1.67 0.05 C302 C303

DFIX 1.54 0.05 C270 C269 C270 C268 C270 C267

DFIX 2.5 0.05 C268 C269 C267 C268 C269 C267

DFIX 1.51 0.05 C083 C288 C083 C144

DFIX 1.51 C172 C155 C155 C260 C260 C259 C259 C257 C257 C109 C109 C229

DFIX 1.51 C155 C247 C247 C261 C261 C161 C161 C116 C116 C230

DFIX 1.51 C109 C322 C322 C321 C321 C323 C323 C091 C091 C226

DFIX 1.51 C091 C324 C324 C325 C161 C325

DFIX 2.55 C091 C325 C324 C161

DFIX 1.51 C082 C135

SIMU 0.01 0.02 2

RIGU

L.S. 12
PLAN 100
SIZE 0.2 0.2 0.2
TEMP -173
BOND \$H
CONF
BUMP 0.02000
ABIN
fmap 2
acta
MERG 2
BASF 0.0044
SHEL 999 0.9
TWIN 0 1 0 1 0 0 0 0 -1 2
WGHT 0.1692 2.4027
FVAR 0.41804 0.48813

Zn01 4 0.83798 0.04062 0.19072 11.00000 0.11545 0.13303 0.14755 =
0.02981 0.00481 0.00689
Zn02 4 0.54434 0.14036 0.10651 11.00000 0.13545 0.13074 0.12898 =
-0.00303 -0.02418 -0.00586
Zn03 4 0.69448 0.45080 0.16685 11.00000 0.14042 0.12024 0.12367 =
-0.01818 0.00869 0.00842
Zn04 4 0.59650 0.00631 0.19323 11.00000 0.14052 0.13025 0.13463 =
0.02482 -0.01914 -0.01711
Zn05 4 0.51978 0.19464 0.17878 11.00000 0.12253 0.14060 0.12695 =
0.00533 -0.02271 -0.00525
Zn06 4 0.64110 0.40353 0.23342 11.00000 0.15300 0.12601 0.12795 =
-0.00673 0.01850 0.00605
Zn07 4 0.93765 0.09619 0.08481 11.00000 0.12680 0.13553 0.15832 =
0.02547 0.03312 0.01554
Zn08 4 0.95092 0.26038 0.15032 11.00000 0.12019 0.14106 0.14740 =
0.01722 0.02331 0.00050
Zn09 4 0.50604 0.37654 0.14578 11.00000 0.12098 0.16200 0.12898 =
-0.01451 -0.00498 0.01095
Zn10 4 0.90347 0.33697 0.09771 11.00000 0.13782 0.13072 0.14038 =
0.01338 0.02147 -0.00255
Zn11 4 0.87771 0.44007 0.20350 11.00000 0.16165 0.12508 0.14290 =
-0.02651 0.02022 -0.00631
Zn12 4 0.76061 0.01657 0.13226 11.00000 0.13594 0.11079 0.14487 =
0.02488 0.01153 0.00283
Cl13 5 0.56809 0.02391 0.24319 11.00000 0.16382 0.16294 0.11851 =
0.03274 -0.00757 -0.00608
Cl14 5 0.56228 0.15854 0.21468 11.00000 0.19030 0.17441 0.15333 =
0.02933 -0.04840 0.01474
Cl15 5 0.52186 0.41959 0.10187 11.00000 0.17102 0.17734 0.13135 =
-0.00547 -0.00687 -0.02188
Cl16 5 0.91474 0.40275 0.24375 11.00000 0.19092 0.15104 0.16020 =
-0.02611 0.00964 0.02083
Cl17 5 0.90000 0.07100 0.03949 11.00000 0.17534 0.14919 0.16816 =
0.00264 0.06150 -0.00941
Cl18 5 0.77982 0.06400 0.09184 11.00000 0.26801 0.31501 0.22492 =
0.13146 0.11164 0.12263

CI19 5 0.24027 0.28624 0.03526 11.00000 0.21545 0.22021 0.28663 =
0.00572 0.01228 -0.00180

CI20 5 0.93319 0.26977 0.20111 11.00000 0.36732 0.22098 0.19346 =
0.01437 0.08970 -0.02882

CI21 5 0.33225 0.32249 0.05207 11.00000 0.18425 0.29011 0.25549 =
0.08285 0.00225 0.00888

CI22 5 0.23385 0.35743 -0.01698 11.00000 0.19283 0.29754 0.30693 =
0.09409 0.02671 0.03255

CI23 5 0.65989 0.43612 0.12088 11.00000 0.20404 0.48566 0.17728 =
-0.15813 -0.04931 0.04345

CI24 5 0.26260 0.39261 0.05137 11.00000 0.21168 0.23345 0.37642 =
-0.00170 0.10129 -0.03240

O029 3 0.85773 0.29339 0.10741 11.00000 0.12046 0.13574 0.16587 =
0.01800 0.01712 0.00373

O030 3 0.58249 0.36510 0.23444 11.00000 0.12081 0.14853 0.12238 =
-0.01931 0.00472 -0.00564

O031 3 0.72520 0.39570 0.18402 11.00000 0.19725 0.10209 0.16087 =
0.02178 0.04796 0.02547

O032 3 0.55922 0.23138 0.15104 11.00000 0.11316 0.14272 0.12414 =
-0.00540 -0.02378 -0.00510

O033 3 0.95413 0.32612 0.12985 11.00000 0.12531 0.11889 0.14093 =
0.01066 0.02084 0.00623

O034 3 0.62246 0.06346 0.17070 11.00000 0.16106 0.15382 0.14148 =
0.02638 -0.01100 -0.00731

O035 3 0.82523 0.00460 0.14832 11.00000 0.14708 0.11720 0.15640 =
0.01710 -0.00355 0.01721

O036 3 0.96440 0.15615 0.07123 11.00000 0.12132 0.13731 0.18781 =
0.03179 0.03817 -0.00184

O037 3 0.50818 0.13644 0.14731 11.00000 0.14143 0.13537 0.16092 =
0.00240 -0.01930 -0.00944

O038 3 0.63737 0.45110 0.20048 11.00000 0.13984 0.12383 0.12809 =
-0.03135 0.00356 0.00500

O039 3 0.89521 0.07732 0.18535 11.00000 0.10213 0.14745 0.13816 =
0.01958 0.02098 -0.02774

O040 3 0.59020 0.18392 0.11313 11.00000 0.15154 0.12764 0.14772 =
0.00899 -0.00693 -0.00537

O041 3 0.68029 0.35879 0.21452 11.00000 0.17888 0.13510 0.12915 =
0.00324 0.02864 0.01576

O042 3 0.73564 0.05636 0.16640 11.00000 0.09967 0.12252 0.14342 =
0.02222 -0.01418 -0.00208

O043 3 0.89855 0.23578 0.12649 11.00000 0.13441 0.13696 0.17089 =
0.02270 0.00103 0.00722

O044 3 0.86675 0.39611 0.10715 11.00000 0.14278 0.11971 0.13323 =
-0.00911 0.02772 -0.00431

O045 3 0.84515 0.39493 0.17468 11.00000 0.20090 0.15446 0.16363 =
-0.04417 0.01789 -0.04843

O047 3 0.58278 0.08200 0.10075 11.00000 0.11720 0.13493 0.14423 =
-0.00879 -0.02084 0.01102

O048 3 0.89115 0.11928 0.11893 11.00000 0.16674 0.13026 0.17707 =
0.01396 0.05822 0.00743

O049 3 0.56140 0.34676 0.16309 11.00000 0.18556 0.16880 0.18512 =
-0.03708 -0.05101 0.03812
O050 3 0.82269 0.46573 0.22732 11.00000 0.14829 0.11701 0.16160 =
-0.00656 0.03464 0.01324
O051 3 0.79370 0.08763 0.19390 11.00000 0.12047 0.12821 0.15683 =
0.02245 0.00287 0.00175
O052 3 0.47629 0.32149 0.12606 11.00000 0.11795 0.16004 0.12225 =
-0.02382 -0.00791 0.00241
O053 3 0.65645 -0.02326 0.20321 11.00000 0.13852 0.17019 0.12601 =
0.01838 -0.02294 0.00539
N054 2 0.51461 0.11442 0.06479 11.00000 0.14099 0.13920 0.15157 =
-0.01770 -0.03985 0.00757
N055 2 0.75819 0.47964 0.14861 11.00000 0.14262 0.12376 0.11580 =
0.00775 0.01716 -0.01449
N056 2 0.88746 0.50987 0.19583 11.00000 0.17707 0.12762 0.14430 =
-0.00074 0.03421 -0.01296
N057 2 0.58700 -0.06096 0.17283 11.00000 0.16980 0.12481 0.16986 =
0.01750 -0.03667 -0.02867
N058 2 0.48806 0.19334 0.08800 11.00000 0.10756 0.12511 0.12945 =
-0.00424 -0.01317 -0.00715
N059 2 0.69344 -0.00691 0.11685 11.00000 0.13493 0.13303 0.16655 =
-0.01020 -0.02432 -0.01137
N061 2 1.00724 0.08529 0.08799 11.00000 0.11837 0.15566 0.20581 =
0.04563 0.03703 0.02311
N062 2 0.44421 0.37506 0.16742 11.00000 0.12990 0.16654 0.15570 =
-0.01094 -0.01420 0.01086
N063 2 0.94806 0.03305 0.11316 11.00000 0.12608 0.13579 0.18305 =
0.04135 0.02689 0.01806
N065 2 0.70087 0.51390 0.18890 11.00000 0.14016 0.10247 0.12061 =
-0.00229 0.02092 -0.01203
N067 2 1.01339 0.26476 0.12910 11.00000 0.11235 0.14489 0.13215 =
0.02086 0.00826 -0.00147
N068 2 0.69730 0.45049 0.26204 11.00000 0.15295 0.13153 0.10138 =
-0.00940 0.01855 -0.00353
N069 2 0.50931 0.42643 0.18551 11.00000 0.13643 0.13415 0.14926 =
-0.03415 0.00072 0.01237
N070 2 0.95158 0.29073 0.06081 11.00000 0.11560 0.13733 0.13064 =
0.01694 0.00932 -0.00391
N071 2 0.61711 0.41960 0.27961 11.00000 0.15359 0.16312 0.12728 =
-0.01160 0.02536 -0.01452
N072 2 0.92097 0.37632 0.05816 11.00000 0.12991 0.14695 0.14052 =
0.02015 0.03160 0.02000
N073 2 0.76348 -0.05381 0.13933 11.00000 0.14093 0.11367 0.15312 =
0.00432 0.00038 -0.00098
N074 2 0.79265 -0.01730 0.21687 11.00000 0.11048 0.12969 0.17573 =
0.01386 -0.00677 -0.00473
N075 2 0.93007 0.44845 0.16578 11.00000 0.16107 0.11876 0.13706 =
-0.00209 0.02158 -0.01270
C075 1 0.95303 0.41588 0.15193 11.00000 0.15186 0.13737 0.14525 =
-0.01549 0.01123 -0.01482
AFIX 43
H075 6 0.95048 0.38741 0.16205 11.00000 -1.20000
AFIX 0
N077 2 0.87870 0.01003 0.22540 11.00000 0.11162 0.16081 0.15477 =
0.04920 0.00560 0.00296

N078 2 0.44900 0.19425 0.17069 11.00000 0.11675 0.14995 0.13790 =
0.01418 -0.01332 -0.00860
C080 1 0.97190 0.32134 0.04002 11.00000 0.10804 0.12867 0.12106 =
0.02058 0.01179 0.01153
C081 1 0.68907 0.50611 0.30165 11.00000 0.15136 0.13349 0.08723 =
-0.02434 0.01868 -0.01418
AFIX 43
H081 6 0.67317 0.51681 0.32047 11.00000 -1.20000
AFIX 0
C082 1 0.74045 -0.07333 0.22206 11.00000 0.12198 0.13025 0.18183 =
0.02872 -0.01085 -0.00983
C083 1 0.98014 0.41859 0.12454 11.00000 0.14681 0.13525 0.14460 =
-0.01561 0.00968 -0.01173
C085 1 0.72647 0.52776 0.29086 11.00000 0.15104 0.12804 0.10569 =
-0.00116 0.01143 -0.00750
AFIX 43
H085 6 0.73559 0.55465 0.30129 11.00000 -1.20000
AFIX 0
C087 1 1.00914 0.31138 0.02130 11.00000 0.08667 0.13325 0.13183 =
0.01746 0.01808 0.01388
AFIX 43
H087 6 1.02112 0.33149 0.00538 11.00000 -1.20000
AFIX 0
C090 1 0.47268 0.23093 0.10327 11.00000 0.13260 0.12757 0.13138 =
0.00233 -0.00960 -0.00814
AFIX 43
H090 6 0.49138 0.24648 0.11840 11.00000 -1.20000
AFIX 0
C091 1 0.73947 0.31964 0.18511 11.00000 0.19860 0.15770 0.19614 =
0.01515 0.02475 0.01654
PART 1
AFIX 23
H09A 6 0.75052 0.31149 0.20753 21.00000 -1.20000
H09B 6 0.71438 0.29889 0.18124 21.00000 -1.20000
AFIX 0
PART 0
PART 2
AFIX 23
H09C 6 0.77134 0.32358 0.19068 -21.00000 -1.20000
H09D 6 0.72790 0.29372 0.19766 -21.00000 -1.20000
AFIX 0
PART 0
C094 1 1.14814 0.14554 0.03837 11.00000 0.13574 0.18289 0.20982 =
0.02534 0.04727 0.01104
C095 1 0.73084 0.53707 0.18239 11.00000 0.14261 0.09542 0.13570 =
0.00941 0.01830 -0.02464
AFIX 43
H095 6 0.73230 0.56614 0.19166 11.00000 -1.20000
AFIX 0
C099 1 0.54285 0.45012 0.19331 11.00000 0.13135 0.13634 0.14329 =
-0.01283 -0.00168 0.01644
AFIX 43
H099 6 0.56691 0.45001 0.17795 11.00000 -1.20000
AFIX 0
C100 1 0.67537 0.47055 0.28583 11.00000 0.13869 0.12583 0.08866 =

-0.01572 0.00404 -0.01033
C102 1 0.66126 0.01266 0.10143 11.00000 0.14403 0.13984 0.14784 =
-0.02218 -0.02945 -0.00661
AFIX 43
H102 6 0.66376 0.04384 0.09791 11.00000 -1.20000
AFIX 0
C105 1 0.42791 0.24754 0.09665 11.00000 0.12285 0.14183 0.13213 =
0.00477 -0.02395 0.00334
C109 1 0.62886 0.25127 0.12357 11.00000 0.16216 0.14696 0.16645 =
-0.00488 0.03519 -0.00522
PART 1
AFIX 23
H10A 6 0.62967 0.27664 0.13930 21.00000 -1.20000
H10B 6 0.62767 0.26270 0.10046 21.00000 -1.20000
AFIX 0
PART 0
PART 2
AFIX 23
H10C 6 0.63723 0.24178 0.10080 -21.00000 -1.20000
H10D 6 0.61315 0.27993 0.12036 -21.00000 -1.20000
AFIX 0
PART 0
C111 1 0.75129 0.51022 0.26442 11.00000 0.15088 0.12631 0.12705 =
-0.00230 0.01580 0.00938
C113 1 1.12687 0.41157 0.06032 11.00000 0.15157 0.18702 0.21177 =
0.04503 0.03049 -0.03182
AFIX 33
H11A 6 1.10070 0.41951 0.04701 11.00000 -1.50000
H11B 6 1.12939 0.43196 0.07928 11.00000 -1.50000
H11C 6 1.15368 0.41383 0.04650 11.00000 -1.50000
AFIX 0

C115 1 0.29547 0.35527 0.02812 11.00000 0.18442 0.26061 0.29277 =
0.04354 0.03223 0.00801
AFIX 13
H115 6 0.31380 0.37321 0.01223 11.00000 -1.20000
AFIX 0

C116 1 0.82371 0.22386 0.11721 11.00000 0.13642 0.15073 0.19282 =
0.02176 0.00791 0.00245
AFIX 23
H11D 6 0.83073 0.19459 0.10715 11.00000 -1.20000
H11E 6 0.79932 0.23781 0.10431 11.00000 -1.20000
AFIX 0
C118 1 0.44066 0.40244 0.19439 11.00000 0.13386 0.16664 0.16673 =
-0.01321 -0.01320 0.01114
AFIX 43
H118 6 0.41214 0.40474 0.20450 11.00000 -1.20000
AFIX 0
C121 1 0.55400 -0.06471 0.15274 11.00000 0.18230 0.14427 0.14779 =
0.03062 -0.04777 -0.03387
AFIX 43
H121 6 0.54764 -0.09325 0.14371 11.00000 -1.20000
AFIX 0
C124 1 0.62339 -0.00825 0.08963 11.00000 0.15042 0.13614 0.16500 =

-0.01872 -0.03284 -0.00697
C128 1 1.03815 0.22866 0.12912 11.00000 0.09877 0.15168 0.14855 =
0.01759 -0.00144 0.00008
AFIX 43
H128 6 1.06719 0.22907 0.11973 11.00000 -1.20000
AFIX 0
C132 1 0.76548 0.52207 0.16051 11.00000 0.13562 0.11304 0.14928 =
0.00957 0.00784 -0.01864
C134 1 0.51471 0.47257 0.24472 11.00000 0.14362 0.15764 0.13329 =
-0.02398 -0.00229 0.00365
AFIX 43
H134 6 0.51780 0.48430 0.26657 11.00000 -1.20000
AFIX 0
C140 1 0.65796 -0.21339 0.18085 11.00000 0.23867 0.15361 0.26924 =
0.02235 -0.07289 -0.01235
C141 1 0.54873 0.47718 0.22276 11.00000 0.13768 0.14201 0.13449 =
-0.01868 -0.00713 0.01143
C143 1 0.95356 0.36395 0.03860 11.00000 0.12333 0.13147 0.12592 =
0.01107 0.01889 0.00731
AFIX 43
H143 6 0.96508 0.38466 0.02293 11.00000 -1.20000
AFIX 0
C144 1 0.98232 0.46375 0.11125 11.00000 0.16105 0.14305 0.15876 =
-0.00478 0.03248 -0.00992
AFIX 43
H144 6 0.99847 0.47030 0.09151 11.00000 -1.20000
AFIX 0
C145 1 1.02677 0.27234 0.02701 11.00000 0.11468 0.13848 0.14650 =
0.02390 0.02115 0.00758
AFIX 43
H145 6 1.05538 0.26692 0.01776 11.00000 -1.20000
AFIX 0
C149 1 0.46434 0.17339 0.06673 11.00000 0.13271 0.12189 0.12582 =
-0.00498 -0.03769 -0.00338
C152 1 1.02167 0.05076 0.10442 11.00000 0.12588 0.16277 0.20602 =
0.04062 0.02233 0.02081
AFIX 43
H152 6 1.05243 0.04341 0.10486 11.00000 -1.20000
AFIX 0
C153 1 0.76728 -0.09731 0.24358 11.00000 0.12042 0.13516 0.17872 =
0.02909 -0.01970 -0.01193
AFIX 43
H153 6 0.75737 -0.12418 0.25362 11.00000 -1.20000
AFIX 0
C154 1 0.73518 0.46851 0.25131 11.00000 0.13870 0.12849 0.11908 =
-0.00664 0.01565 0.01416
AFIX 43
H154 6 0.75197 0.45356 0.23466 11.00000 -1.20000
AFIX 0
C155 1 0.72544 0.12947 0.19104 11.00000 0.13185 0.11297 0.15329 =
0.02119 0.00772 -0.00313
PART 1
AFIX 23
H15A 6 0.74324 0.15002 0.20498 21.00000 -1.20000
H15B 6 0.69820 0.12118 0.20346 21.00000 -1.20000

AFIX 0
PART 0
PART 2
AFIX 23
H15C 6 0.71304 0.12713 0.21382 -21.00000 -1.20000
H15D 6 0.70048 0.13300 0.17514 -21.00000 -1.20000
AFIX 0
PART 0
C157 1 1.17617 0.11856 0.06227 11.00000 0.13663 0.24412 0.23787 =
0.03378 0.01042 0.00148
AFIX 33
H15E 6 1.16061 0.09083 0.06778 11.00000 -1.50000
H15F 6 1.20485 0.11157 0.05187 11.00000 -1.50000
H15G 6 1.18121 0.13578 0.08266 11.00000 -1.50000
AFIX 0
C158 1 0.96378 -0.03113 0.15899 11.00000 0.13662 0.14591 0.19316 =
0.04964 0.01282 0.02972
AFIX 43
H158 6 0.96989 -0.05138 0.17658 11.00000 -1.20000
AFIX 0
C160 1 0.52551 -0.07205 -0.00335 11.00000 0.28454 0.19886 0.25346 =
-0.04124 -0.10261 0.04068
AFIX 33
H16A 6 0.55345 -0.08203 0.00683 11.00000 -1.50000
H16B 6 0.50089 -0.07694 0.01225 11.00000 -1.50000
H16C 6 0.52010 -0.08896 -0.02383 11.00000 -1.50000
AFIX 0
C162 1 0.82600 0.52847 0.12526 11.00000 0.16005 0.12660 0.16700 =
0.01484 0.03857 -0.00871
AFIX 43
H162 6 0.84964 0.54636 0.11700 11.00000 -1.20000
AFIX 0
C163 1 0.47789 0.43118 0.35985 11.00000 0.28784 0.31137 0.22501 =
-0.04006 0.13153 -0.01510
AFIX 33
H16D 6 0.50166 0.45278 0.35553 11.00000 -1.50000
H16E 6 0.45534 0.43329 0.34227 11.00000 -1.50000
H16F 6 0.46413 0.43761 0.38145 11.00000 -1.50000
AFIX 0
C164 1 0.86745 0.67523 0.22399 11.00000 0.25984 0.12962 0.22315 =
-0.01021 0.00735 -0.01790
AFIX 33
H16G 6 0.89665 0.66072 0.22498 11.00000 -1.50000
H16H 6 0.85503 0.67198 0.20155 11.00000 -1.50000
H16I 6 0.87072 0.70693 0.22926 11.00000 -1.50000
AFIX 0
C165 1 0.47688 0.45186 0.23628 11.00000 0.14543 0.16708 0.14407 =
-0.02417 -0.00102 0.00227
AFIX 43
H165 6 0.45158 0.45489 0.25030 11.00000 -1.20000
AFIX 0
C166 1 0.62334 -0.05223 0.09108 11.00000 0.15086 0.13332 0.17597 =
-0.01237 -0.03934 -0.01464
AFIX 43
H166 6 0.60039 -0.06867 0.08040 11.00000 -1.20000

AFIX 0
C168 1 0.96705 0.24896 0.06342 11.00000 0.10693 0.13274 0.14643 =
0.02816 0.01186 -0.01546
AFIX 43
H168 6 0.95222 0.22722 0.07655 11.00000 -1.20000
AFIX 0
C169 1 0.48113 0.13419 0.05277 11.00000 0.13807 0.12564 0.14581 =
-0.00225 -0.03766 0.00166
AFIX 43
H169 6 0.46694 0.12206 0.03367 11.00000 -1.20000
AFIX 0
C172 1 0.75229 0.08831 0.18267 11.00000 0.12377 0.11818 0.14444 =
0.02391 0.00878 -0.00782
C173 1 0.68875 -0.05417 0.11794 11.00000 0.14206 0.11882 0.16875 =
-0.01179 -0.02164 -0.02330
C177 1 0.40523 0.22647 0.07092 11.00000 0.13353 0.14284 0.12928 =
-0.00251 -0.02179 0.01353
AFIX 43
H177 6 0.37672 0.23698 0.06434 11.00000 -1.20000
AFIX 0
C180 1 1.00796 0.23842 0.04505 11.00000 0.11978 0.14414 0.15081 =
0.01639 0.01958 -0.00582
C186 1 0.42428 0.18958 0.05451 11.00000 0.13657 0.13155 0.13231 =
-0.00633 -0.02188 -0.00298
AFIX 43
H186 6 0.41021 0.17632 0.03579 11.00000 -1.20000
AFIX 0
AFIX 66
C101 1 1.00746 0.15550 0.06493 11.00000 0.11381 0.14145 0.19023 =
0.01919 0.04366 0.01352
C127 1 1.02951 0.19266 0.05252 11.00000 0.11958 0.14951 0.17257 =
0.02122 0.03891 -0.00224
C104 1 1.07479 0.19046 0.04554 11.00000 0.12508 0.14834 0.18982 =
0.04094 0.03858 0.00101
AFIX 43
H104 6 1.08987 0.21585 0.03706 11.00000 -1.20000
AFIX 0
AFIX 65
C112 1 1.09803 0.15109 0.05097 11.00000 0.12649 0.14969 0.20474 =
0.02766 0.03262 0.01107
C147 1 1.07598 0.11392 0.06338 11.00000 0.12631 0.13145 0.20956 =
0.02348 0.03538 0.02770
AFIX 43
H147 6 1.09186 0.08701 0.06710 11.00000 -1.20000
AFIX 0
AFIX 65
C187 1 1.03069 0.11613 0.07036 11.00000 0.12571 0.13776 0.20977 =
0.02274 0.03656 0.01690
AFIX 0
C188 1 0.65525 -0.07335 0.10739 11.00000 0.15077 0.12941 0.17505 =
-0.01443 -0.03111 -0.01916
AFIX 43
H188 6 0.65227 -0.10445 0.11113 11.00000 -1.20000
AFIX 0
AFIX 66

N060 2 0.53619 0.00663 0.16326 11.00000 0.16908 0.16812 0.13858 =
0.03421 -0.04996 -0.02361
C079 1 0.52857 -0.03107 0.14400 11.00000 0.17921 0.15593 0.15178 =
0.02642 -0.04149 -0.02495
C139 1 0.49576 -0.03056 0.11959 11.00000 0.17633 0.14933 0.15312 =
0.01009 -0.05426 -0.02389
AFIX 43
H139 6 0.49056 -0.05633 0.10642 11.00000 -1.20000
AFIX 0
AFIX 65
C185 1 0.47057 0.00766 0.11443 11.00000 0.18355 0.15260 0.14992 =
0.01796 -0.05468 -0.01693
AFIX 43
H185 6 0.44815 0.00801 0.09774 11.00000 -1.20000
AFIX 0
AFIX 65
C189 1 0.47819 0.04536 0.13369 11.00000 0.16046 0.14747 0.13498 =
0.02899 -0.06042 -0.01927
C193 1 0.51099 0.04485 0.15810 11.00000 0.15495 0.14693 0.12671 =
0.04802 -0.05277 -0.03249
AFIX 43
H193 6 0.51620 0.07061 0.17127 11.00000 -1.20000
AFIX 0
C194 1 0.75096 -0.03178 0.21166 11.00000 0.11544 0.12093 0.17978 =
0.01285 -0.00676 -0.00829
AFIX 43
H194 6 0.72950 -0.01344 0.20099 11.00000 -1.20000
AFIX 0
AFIX 66
C084 1 0.83808 -0.04064 0.14949 11.00000 0.14286 0.11374 0.16299 =
0.02376 0.00580 0.01320
C195 1 0.80354 -0.07145 0.14967 11.00000 0.14444 0.11382 0.16159 =
0.01989 0.01219 0.00510
C142 1 0.81209 -0.11572 0.15750 11.00000 0.15864 0.11741 0.18024 =
0.03057 0.01573 0.00189
AFIX 43
H142 6 0.78848 -0.13678 0.15761 11.00000 -1.20000
AFIX 0
AFIX 65
C117 1 0.85518 -0.12918 0.16515 11.00000 0.17736 0.13858 0.21134 =
0.03198 -0.00012 0.00578
C108 1 0.88972 -0.09837 0.16498 11.00000 0.16549 0.12407 0.19685 =
0.03784 -0.00862 0.01502
AFIX 43
H108 6 0.91917 -0.10757 0.17021 11.00000 -1.20000
AFIX 0
AFIX 65
C148 1 0.88118 -0.05410 0.15715 11.00000 0.14872 0.12358 0.17872 =
0.02899 -0.00338 0.01657
AFIX 0
C196 1 0.91273 0.52085 0.17430 11.00000 0.17730 0.13921 0.16319 =
0.00491 0.04135 -0.00474
AFIX 43
H196 6 0.91769 0.55179 0.17107 11.00000 -1.20000
AFIX 0

C197 1 0.91983 -0.02569 0.14811 11.00000 0.14014 0.11661 0.18373 =
0.03782 0.00165 0.02263
C199 1 0.48918 -0.00306 -0.02702 11.00000 0.27981 0.23017 0.21343 =
-0.02597 -0.10932 0.00524
AFIX 33
H19A 6 0.49517 0.02855 -0.03113 11.00000 -1.50000
H19B 6 0.48272 -0.01798 -0.04818 11.00000 -1.50000
H19C 6 0.46352 -0.00596 -0.01211 11.00000 -1.50000
AFIX 0
C202 1 1.12385 0.33669 0.03982 11.00000 0.15958 0.23126 0.18710 =
-0.00086 0.07207 -0.02975
AFIX 33
H20A 6 1.09835 0.34455 0.02576 11.00000 -1.50000
H20B 6 1.15155 0.34339 0.02794 11.00000 -1.50000
H20C 6 1.12277 0.30484 0.04507 11.00000 -1.50000
AFIX 0
AFIX 66
C131 1 0.79198 0.53157 0.24995 11.00000 0.15792 0.12475 0.14558 =
-0.01508 0.01765 0.00345
C136 1 0.79668 0.57684 0.25626 11.00000 0.18175 0.12325 0.16568 =
-0.01430 0.00800 -0.00189
AFIX 43
H136 6 0.77554 0.59183 0.26982 11.00000 -1.20000
AFIX 0
AFIX 65
C159 1 0.83231 0.60018 0.24271 11.00000 0.19951 0.13510 0.18560 =
-0.02307 0.01339 -0.00916
C174 1 0.86324 0.57825 0.22285 11.00000 0.19384 0.13515 0.17579 =
-0.01825 0.01220 -0.00563
AFIX 43
H174 6 0.88760 0.59420 0.21359 11.00000 -1.20000
AFIX 0
AFIX 65
C179 1 0.85855 0.53298 0.21655 11.00000 0.18102 0.12710 0.15616 =
-0.01392 0.02155 -0.01425
C203 1 0.82292 0.50964 0.23010 11.00000 0.16393 0.12185 0.15187 =
-0.01628 0.02372 -0.00309
AFIX 0
C204 1 0.30425 0.06933 0.12995 11.00000 0.16112 0.25330 0.29618 =
-0.00487 -0.03625 -0.06565
AFIX 33
H20D 6 0.32352 0.04390 0.12482 11.00000 -1.50000
H20E 6 0.30244 0.07317 0.15415 11.00000 -1.50000
H20F 6 0.27440 0.06391 0.12097 11.00000 -1.50000
AFIX 0
C205 1 0.47347 0.42628 0.20791 11.00000 0.13360 0.15237 0.15377 =
-0.02608 0.00040 0.00261
AFIX 9
C089 1 0.58429 0.54508 0.24415 11.00000 0.14452 0.12644 0.15553 =
-0.02529 0.01337 0.01313
AFIX 43
H089 6 0.55766 0.55173 0.25592 11.00000 -1.20000
AFIX 0
AFIX 5
C092 1 0.58933 0.50359 0.22853 11.00000 0.14124 0.13248 0.13500 =

-0.02554 0.00957 0.01109
C125 1 0.62864 0.49383 0.21112 11.00000 0.13833 0.12479 0.12274 =
-0.02678 0.00554 -0.00332
C129 1 0.66288 0.52547 0.20937 11.00000 0.12998 0.11398 0.12257 =
-0.01470 0.00433 -0.00078
C133 1 0.65785 0.56693 0.22500 11.00000 0.13049 0.12158 0.14638 =
-0.02021 -0.00315 0.00563
AFIX 43
H133 6 0.68105 0.58838 0.22380 11.00000 -1.20000
AFIX 0
AFIX 5
C206 1 0.61853 0.57673 0.24240 11.00000 0.14758 0.12263 0.16455 =
-0.01329 0.00738 0.02053
AFIX 0
C208 1 0.99835 -0.00779 0.14476 11.00000 0.13003 0.15086 0.19934 =
0.04846 0.01533 0.02730
AFIX 43
H208 6 1.02833 -0.01385 0.15079 11.00000 -1.20000
AFIX 0
C210 1 0.86120 -0.17745 0.17907 11.00000 0.21663 0.14453 0.25473 =
0.04580 0.00405 0.01069
C211 1 0.93639 0.48983 0.15269 11.00000 0.16269 0.12516 0.15590 =
0.01073 0.04101 -0.01021
AFIX 66
C106 1 0.69609 -0.13853 0.20498 11.00000 0.16216 0.15094 0.22425 =
0.04616 -0.03506 -0.01544
AFIX 43
H106 6 0.72103 -0.15564 0.21193 11.00000 -1.20000
AFIX 0
AFIX 65
C135 1 0.69576 -0.09270 0.20983 11.00000 0.13963 0.14157 0.18976 =
0.04045 -0.02489 -0.01809
C191 1 0.65926 -0.06767 0.19967 11.00000 0.14077 0.15177 0.16265 =
0.03369 -0.02347 -0.01603
C198 1 0.62310 -0.08848 0.18467 11.00000 0.15445 0.14790 0.18101 =
0.02338 -0.03694 -0.01009
C212 1 0.62344 -0.13431 0.17983 11.00000 0.16820 0.14952 0.21871 =
0.03229 -0.04649 -0.02116
AFIX 43
H212 6 0.59873 -0.14853 0.16957 11.00000 -1.20000
AFIX 0
AFIX 65
C200 1 0.65993 -0.15934 0.18998 11.00000 0.19026 0.15081 0.23926 =
0.03696 -0.05788 -0.01320
AFIX 0
C213 1 0.72660 -0.07425 0.13158 11.00000 0.15046 0.12232 0.17134 =
-0.00130 -0.00928 -0.00782
AFIX 43
H213 6 0.72533 -0.10541 0.13554 11.00000 -1.20000
AFIX 0
C214 1 0.78797 0.46435 0.12630 11.00000 0.15571 0.12908 0.12316 =
0.02430 0.02760 -0.00060
AFIX 43
H214 6 0.78311 0.43464 0.11903 11.00000 -1.20000
AFIX 0

C215 1 1.02337 -0.01093 0.29486 11.00000 0.11937 0.30016 0.22614 =
0.10135 -0.02655 0.02187
AFIX 33
H21A 6 0.99443 -0.02599 0.29431 11.00000 -1.50000
H21B 6 1.04520 -0.03003 0.30604 11.00000 -1.50000
H21C 6 1.02057 0.01722 0.30704 11.00000 -1.50000
AFIX 0
C216 1 0.29086 0.30836 0.15364 11.00000 0.12776 0.23982 0.19932 =
-0.00521 0.00183 -0.00292
C217 1 0.69462 -0.23515 0.19765 11.00000 0.27163 0.15894 0.30176 =
0.00468 -0.10512 -0.00020
AFIX 33
H21D 6 0.72277 -0.22169 0.19050 11.00000 -1.50000
H21E 6 0.69125 -0.23160 0.22178 11.00000 -1.50000
H21F 6 0.69465 -0.26692 0.19204 11.00000 -1.50000
AFIX 0
C218 1 0.32347 0.11114 0.11435 11.00000 0.13534 0.21927 0.23621 =
0.00515 -0.06199 -0.02244
C219 1 0.27085 0.35446 0.14714 11.00000 0.12612 0.25041 0.28113 =
0.00339 -0.01144 0.00999
AFIX 33
H21G 6 0.27492 0.36235 0.12365 11.00000 -1.50000
H21H 6 0.23899 0.35400 0.15242 11.00000 -1.50000
H21I 6 0.28584 0.37654 0.16119 11.00000 -1.50000
AFIX 0
C220 1 1.14620 0.11692 0.00645 11.00000 0.17437 0.26674 0.20993 =
-0.00692 0.04688 0.00435
AFIX 33
H22A 6 1.13471 0.08726 0.01191 11.00000 -1.50000
H22B 6 1.12650 0.13117 -0.00983 11.00000 -1.50000
H22C 6 1.17617 0.11416 -0.00299 11.00000 -1.50000
AFIX 0
C221 1 0.83580 0.65324 0.24952 11.00000 0.24503 0.14394 0.21540 =
-0.02958 0.01548 -0.01944
C222 1 1.16610 0.18960 0.02670 11.00000 0.13986 0.21741 0.28382 =
0.03613 0.06798 0.00511
AFIX 33
H22D 6 1.14473 0.20344 0.01144 11.00000 -1.50000
H22E 6 1.17082 0.20914 0.04593 11.00000 -1.50000
H22F 6 1.19446 0.18493 0.01514 11.00000 -1.50000
AFIX 0
C223 1 0.79856 0.54729 0.14752 11.00000 0.14618 0.12036 0.17164 =
-0.00141 0.02485 -0.01432
AFIX 43
H223 6 0.80222 0.57749 0.15400 11.00000 -1.20000
AFIX 0
C224 1 0.26653 0.27359 0.13153 11.00000 0.14199 0.26757 0.23434 =
-0.00476 -0.00748 -0.03040
AFIX 33
H22G 6 0.26926 0.28217 0.10807 11.00000 -1.50000
H22H 6 0.28000 0.24420 0.13486 11.00000 -1.50000
H22I 6 0.23498 0.27246 0.13771 11.00000 -1.50000
AFIX 0
C225 1 0.91378 0.00904 0.12543 11.00000 0.12955 0.11803 0.17315 =
0.03038 0.01449 0.01982

AFIX 43

H225 6 0.88441 0.01603 0.11836 11.00000 -1.20000

AFIX 0

C226 1 0.71251 0.36270 0.19429 11.00000 0.18467 0.13024 0.15684 =
0.03076 0.03536 0.01329

C228 1 0.28586 0.29398 0.18924 11.00000 0.14335 0.31851 0.21878 =
0.01799 0.00069 -0.04819

AFIX 33

H22J 6 0.29911 0.26441 0.19214 11.00000 -1.50000

H22K 6 0.30102 0.31534 0.20380 11.00000 -1.50000

H22L 6 0.25418 0.29280 0.19503 11.00000 -1.50000

AFIX 0

C229 1 0.58954 0.21959 0.13106 11.00000 0.14697 0.14162 0.14405 =
0.00753 0.00989 -0.00891

C230 1 0.86347 0.25277 0.11722 11.00000 0.13143 0.13668 0.16856 =
0.01905 0.00212 0.00421

C232 1 1.03872 -0.00159 0.25950 11.00000 0.14390 0.26094 0.23418 =
0.09300 -0.01448 0.00007

C233 1 0.61359 -0.23215 0.19047 11.00000 0.27152 0.16454 0.33488 =
0.01245 -0.07973 -0.05713

AFIX 33

H23A 6 0.59001 -0.21632 0.17845 11.00000 -1.50000

H23B 6 0.61257 -0.26386 0.18477 11.00000 -1.50000

H23C 6 0.60917 -0.22855 0.21450 11.00000 -1.50000

AFIX 0

C234 1 1.08190 0.02114 0.25396 11.00000 0.12421 0.30309 0.26440 =
0.11380 -0.01323 -0.00629

AFIX 33

H23D 6 1.08697 0.02468 0.22999 11.00000 -1.50000

H23E 6 1.08134 0.05050 0.26460 11.00000 -1.50000

H23F 6 1.10596 0.00326 0.26360 11.00000 -1.50000

AFIX 0

C235 1 0.79282 0.67485 0.24789 11.00000 0.27301 0.13077 0.31417 =
-0.03072 0.04326 0.00498

AFIX 33

H23G 6 0.78180 0.67407 0.22493 11.00000 -1.50000

H23H 6 0.77181 0.65927 0.26248 11.00000 -1.50000

H23I 6 0.79574 0.70587 0.25516 11.00000 -1.50000

AFIX 0

C236 1 1.04614 -0.05168 0.24548 11.00000 0.19476 0.28144 0.27052 =
0.07143 -0.03129 0.01735

AFIX 33

H23J 6 1.01868 -0.06890 0.24834 11.00000 -1.50000

H23K 6 1.05387 -0.05040 0.22179 11.00000 -1.50000

H23L 6 1.07036 -0.06600 0.25785 11.00000 -1.50000

AFIX 0

C237 1 0.82182 -0.20283 0.18466 11.00000 0.25721 0.13071 0.32018 =
0.06564 -0.01416 -0.01140

AFIX 33

H23M 6 0.80250 -0.18700 0.20036 11.00000 -1.50000

H23N 6 0.80605 -0.20702 0.16352 11.00000 -1.50000

H23O 6 0.82983 -0.23193 0.19393 11.00000 -1.50000

AFIX 0

C238 1 0.86813 0.56430 0.00463 11.00000 0.33528 0.20910 0.26751 =
0.11645 0.11599 0.06754

AFIX 33
H23P 6 0.85757 0.57422 0.02649 11.00000 -1.50000
H23Q 6 0.88388 0.58877 -0.00641 11.00000 -1.50000
H23R 6 0.84265 0.55519 -0.00902 11.00000 -1.50000
AFIX 0
AFIX 66
C103 1 0.57153 -0.00915 0.04208 11.00000 0.19662 0.15106 0.18419 =
-0.03369 -0.04179 0.01783
AFIX 43
H103 6 0.58433 -0.03735 0.03726 11.00000 -1.20000
AFIX 0
AFIX 65
C120 1 0.58745 0.01593 0.06863 11.00000 0.16279 0.14333 0.16857 =
-0.02092 -0.03449 0.01175
C175 1 0.56873 0.05720 0.07568 11.00000 0.14954 0.14367 0.15568 =
-0.01159 -0.03556 0.01199
C176 1 0.53409 0.07340 0.05617 11.00000 0.15981 0.15134 0.14139 =
-0.01890 -0.04993 0.01366
C178 1 0.51816 0.04832 0.02962 11.00000 0.19729 0.16422 0.16263 =
-0.02430 -0.06190 0.02678
AFIX 43
H178 6 0.49449 0.05939 0.01630 11.00000 -1.20000
AFIX 0
AFIX 65
C240 1 0.53689 0.00705 0.02257 11.00000 0.21965 0.17448 0.19823 =
-0.03605 -0.05761 0.02502
AFIX 0
C241 1 0.29770 0.15075 0.12034 11.00000 0.15896 0.23899 0.30341 =
-0.02002 -0.05750 -0.02568
AFIX 33
H24A 6 0.29660 0.15670 0.14433 11.00000 -1.50000
H24B 6 0.31157 0.17603 0.10891 11.00000 -1.50000
H24C 6 0.26739 0.14649 0.11188 11.00000 -1.50000
AFIX 0
C242 1 0.56809 -0.01179 -0.03749 11.00000 0.31863 0.25867 0.21953 =
-0.05782 -0.04422 0.02032
AFIX 33
H24D 6 0.59631 -0.02417 -0.02948 11.00000 -1.50000
H24E 6 0.56102 -0.02451 -0.05934 11.00000 -1.50000
H24F 6 0.57073 0.02064 -0.03945 11.00000 -1.50000
AFIX 0
C243 1 0.66274 -0.21507 0.14087 11.00000 0.33490 0.18976 0.28605 =
-0.01893 -0.07603 0.00256
AFIX 33
H24G 6 0.63725 -0.19997 0.13064 11.00000 -1.50000
H24H 6 0.69033 -0.20009 0.13419 11.00000 -1.50000
H24I 6 0.66355 -0.24616 0.13348 11.00000 -1.50000
AFIX 0
C244 1 0.89365 -0.19772 0.15558 11.00000 0.31288 0.14910 0.31843 =
0.03723 0.05365 0.04370
AFIX 33
H24J 6 0.91952 -0.17807 0.15327 11.00000 -1.50000
H24K 6 0.90330 -0.22668 0.16420 11.00000 -1.50000
H24L 6 0.87952 -0.20177 0.13379 11.00000 -1.50000
AFIX 0

C245 1 0.32692 0.09963 0.07898 11.00000 0.14337 0.28351 0.25065 =
-0.01559 -0.06662 -0.02163

AFIX 33

H24M 6 0.34476 0.07250 0.07650 11.00000 -1.50000
H24N 6 0.29707 0.09457 0.06988 11.00000 -1.50000
H24O 6 0.34125 0.12411 0.06691 11.00000 -1.50000

AFIX 0

C246 1 0.26827 0.32317 0.00805 11.00000 0.19020 0.26653 0.28714 =
0.04485 0.02041 0.00641

AFIX 13

H246 6 0.28857 0.30563 -0.00671 11.00000 -1.20000

AFIX 0

C248 1 0.91744 0.50760 -0.02310 11.00000 0.31295 0.21704 0.22876 =
0.09680 0.09590 0.04465

AFIX 33

H24P 6 0.92452 0.53316 -0.03733 11.00000 -1.50000
H24Q 6 0.94433 0.48970 -0.01964 11.00000 -1.50000
H24R 6 0.89455 0.48933 -0.03386 11.00000 -1.50000

AFIX 0

C249 1 0.85695 0.65189 0.28340 11.00000 0.36083 0.14811 0.23238 =
-0.00186 -0.01899 -0.08013

AFIX 33

H24S 6 0.88577 0.63670 0.28198 11.00000 -1.50000
H24T 6 0.86140 0.68235 0.29150 11.00000 -1.50000
H24U 6 0.83747 0.63575 0.29881 11.00000 -1.50000

AFIX 0

C250 1 0.90083 0.52352 0.00917 11.00000 0.26764 0.19214 0.23681 =
0.07675 0.08984 0.03485

C251 1 0.52870 -0.02388 -0.01138 11.00000 0.26178 0.20311 0.21957 =
-0.03785 -0.07554 0.02197

C252 1 0.88518 -0.17276 0.21286 11.00000 0.29829 0.18442 0.28658 =
0.04994 -0.03786 0.02913

AFIX 33

H25A 6 0.91253 -0.15540 0.20991 11.00000 -1.50000
H25B 6 0.86558 -0.15752 0.22870 11.00000 -1.50000
H25C 6 0.89265 -0.20238 0.22144 11.00000 -1.50000

AFIX 0

C253 1 0.94633 0.55007 0.02258 11.00000 0.32500 0.22012 0.29952 =
0.07794 0.07262 -0.01491

AFIX 33

H25D 6 0.94050 0.56300 0.04456 11.00000 -1.50000
H25E 6 0.97112 0.52896 0.02417 11.00000 -1.50000
H25F 6 0.95405 0.57383 0.00688 11.00000 -1.50000

AFIX 0

C161 1 0.80999 0.21822 0.15403 11.00000 0.15812 0.17134 0.20793 =
0.02024 0.01514 0.00655

PART 1

AFIX 23

H16J 6 0.83481 0.20183 0.16466 21.00000 -1.20000
H16K 6 0.78437 0.19747 0.15378 21.00000 -1.20000

AFIX 0

PART 0

PART 2

AFIX 23
H16L 6 0.83560 0.20775 0.16743 -21.00000 -1.20000
H16M 6 0.79956 0.24695 0.16327 -21.00000 -1.20000
AFIX 0
PART 0
AFIX 66
C088 1 0.96179 0.15713 0.17627 11.00000 0.10527 0.14697 0.13740 =
0.02374 -0.00545 -0.00109
AFIX 43
H088 6 0.93289 0.15922 0.18577 11.00000 -1.20000
AFIX 0
AFIX 65
N066 2 0.97850 0.19208 0.15731 11.00000 0.10935 0.13427 0.12442 =
0.01380 0.00769 0.01336
C123 1 1.02078 0.18903 0.14340 11.00000 0.10946 0.14808 0.14625 =
0.01781 0.00122 0.01461
C181 1 1.04635 0.15104 0.14845 11.00000 0.10824 0.15982 0.16233 =
0.03281 0.01511 0.02244
AFIX 43
H181 6 1.07524 0.14896 0.13894 11.00000 -1.20000
AFIX 0
AFIX 65
C182 1 1.02964 0.11610 0.16740 11.00000 0.12527 0.16088 0.17055 =
0.03333 0.00858 0.01597
AFIX 43
H182 6 1.04711 0.09013 0.17085 11.00000 -1.20000
AFIX 0
AFIX 65
C263 1 0.98736 0.11914 0.18131 11.00000 0.11047 0.16165 0.15822 =
0.04175 -0.00185 0.00155
AFIX 0
C264 1 0.80672 -0.08124 0.24938 11.00000 0.11639 0.14494 0.17273 =
0.03215 -0.00162 0.00269
AFIX 43
H264 6 0.82683 -0.09713 0.26324 11.00000 -1.20000
AFIX 0
C265 1 0.98877 0.02368 0.12222 11.00000 0.12773 0.15075 0.19879 =
0.04600 0.01936 0.02323
C267 1 0.56876 0.62220 0.28120 11.00000 0.20675 0.13975 0.22027 =
-0.05007 0.05416 0.00942
AFIX 33
H26A 6 0.54437 0.62284 0.26498 11.00000 -1.50000
H26B 6 0.56578 0.59579 0.29536 11.00000 -1.50000
H26C 6 0.56756 0.64905 0.29504 11.00000 -1.50000
AFIX 0
C268 1 0.61420 0.65907 0.23734 11.00000 0.24604 0.14145 0.20819 =
-0.02108 0.04127 0.03062
AFIX 33
H26D 6 0.58888 0.65664 0.22205 11.00000 -1.50000
H26E 6 0.61295 0.68775 0.24895 11.00000 -1.50000
H26F 6 0.64205 0.65704 0.22471 11.00000 -1.50000
AFIX 0
C269 1 0.64786 0.61964 0.28788 11.00000 0.23714 0.16458 0.21691 =
-0.07850 -0.02091 0.05673
AFIX 33

H26G 6 0.67669 0.61855 0.27648 11.00000 -1.50000
H26H 6 0.64632 0.64650 0.30169 11.00000 -1.50000
H26I 6 0.64455 0.59324 0.30201 11.00000 -1.50000
AFIX 0
C270 1 0.61217 0.62056 0.26326 11.00000 0.17285 0.12437 0.18984 =
-0.02791 0.01518 0.03066
AFIX 66
C093 1 0.99750 0.02857 0.24441 11.00000 0.13443 0.22541 0.20805 =
0.08587 -0.01140 -0.00261
C097 1 1.00597 0.06526 0.22428 11.00000 0.12464 0.20514 0.19551 =
0.07534 -0.00898 -0.00659
AFIX 43
H097 6 1.03478 0.07833 0.22404 11.00000 -1.20000
AFIX 0
AFIX 65
C266 1 0.97229 0.08282 0.20450 11.00000 0.11314 0.17178 0.16967 =
0.05389 -0.00197 -0.00251
C275 1 0.93014 0.06370 0.20485 11.00000 0.11048 0.16077 0.16495 =
0.04367 0.00359 -0.01054
C274 1 0.92167 0.02701 0.22498 11.00000 0.11220 0.17463 0.16785 =
0.05231 0.00678 0.00721
C276 1 0.95535 0.00944 0.24476 11.00000 0.12144 0.20582 0.19145 =
0.06787 -0.00272 0.00692
AFIX 43
H276 6 0.94956 -0.01564 0.25852 11.00000 -1.20000
AFIX 0
C277 1 0.81958 -0.04097 0.23539 11.00000 0.10330 0.14327 0.16559 =
0.02607 0.00332 0.00285
C278 1 0.86337 -0.02466 0.23912 11.00000 0.11547 0.15427 0.15640 =
0.04284 -0.00476 0.00556
AFIX 43
H278 6 0.88295 -0.04109 0.25306 11.00000 -1.20000
AFIX 0
C279 1 0.42636 0.22484 0.18436 11.00000 0.13085 0.16428 0.11775 =
0.00876 -0.00493 -0.00611
AFIX 43
H279 6 0.39502 0.22638 0.18137 11.00000 -1.20000
AFIX 0
AFIX 66
C209 1 0.88804 0.48960 0.03469 11.00000 0.21447 0.15980 0.20141 =
0.05512 0.08335 0.02641
C190 1 0.90732 0.44765 0.03219 11.00000 0.18806 0.15226 0.18112 =
0.03671 0.06867 0.02102
AFIX 43
H190 6 0.92561 0.44058 0.01362 11.00000 -1.20000
AFIX 0
AFIX 65
C272 1 0.89984 0.41604 0.05685 11.00000 0.15286 0.13824 0.16170 =
0.01980 0.04766 0.01804
C281 1 0.87308 0.42638 0.08402 11.00000 0.14568 0.11480 0.15859 =
0.00824 0.04082 -0.00399
C110 1 0.85379 0.46833 0.08652 11.00000 0.15416 0.12679 0.16540 =
0.02194 0.05403 0.01166
C114 1 0.86128 0.49994 0.06186 11.00000 0.18693 0.14507 0.18781 =
0.03891 0.06990 0.02366

AFIX 43
H114 6 0.84810 0.52861 0.06357 11.00000 -1.20000
AFIX 0
AFIX 66
C086 1 0.40895 0.28370 0.11696 11.00000 0.11671 0.16573 0.12803 =
-0.00532 -0.02439 0.00643
C150 1 0.36396 0.27933 0.12443 11.00000 0.12491 0.19167 0.14472 =
-0.01637 -0.01738 -0.00315
AFIX 43
H150 6 0.34703 0.25605 0.11461 11.00000 -1.20000
AFIX 0
AFIX 65
C167 1 0.34374 0.30902 0.14626 11.00000 0.12745 0.20734 0.16918 =
-0.01535 -0.00577 0.00448
C283 1 0.36851 0.34309 0.16062 11.00000 0.12212 0.18821 0.16225 =
-0.00751 -0.01084 0.01721
AFIX 43
H283 6 0.35469 0.36339 0.17554 11.00000 -1.20000
AFIX 0
AFIX 65
C284 1 0.41349 0.34746 0.15315 11.00000 0.11791 0.16468 0.14699 =
-0.00132 -0.01240 0.01587
C285 1 0.43371 0.31776 0.13132 11.00000 0.11568 0.15791 0.13661 =
-0.00316 -0.01233 0.00919
AFIX 0
AFIX 66
C282 1 1.02616 0.30443 0.11589 11.00000 0.12282 0.14488 0.14531 =
0.02443 0.01809 -0.00890
C289 1 1.06722 0.31290 0.10122 11.00000 0.12476 0.14640 0.16041 =
0.01656 0.02145 -0.01223
AFIX 43
H289 6 1.08768 0.28924 0.09769 11.00000 -1.20000
AFIX 0
AFIX 65
C287 1 1.07835 0.35600 0.09172 11.00000 0.13083 0.15022 0.15926 =
0.01190 0.02397 -0.02789
C286 1 1.04841 0.39063 0.09688 11.00000 0.13914 0.14173 0.15981 =
0.00644 0.01918 -0.02503
AFIX 43
H286 6 1.05602 0.42008 0.09038 11.00000 -1.20000
AFIX 0
AFIX 65
C288 1 1.00735 0.38215 0.11155 11.00000 0.14080 0.13837 0.15386 =
0.00036 0.02458 -0.01008
C300 1 0.99622 0.33906 0.12106 11.00000 0.12983 0.14278 0.14701 =
0.01362 0.02591 -0.00444
AFIX 0
C301 1 1.12203 0.36435 0.07291 11.00000 0.14377 0.16921 0.17222 =
0.01947 0.03223 -0.02179
C302 1 0.49911 0.38016 0.36033 11.00000 0.22445 0.27704 0.17781 =
-0.03905 0.07881 -0.05863
AFIX 66
C207 1 0.50041 0.34878 0.29948 11.00000 0.16952 0.21394 0.14938 =
-0.02314 0.05040 -0.02748
AFIX 43

H207 6 0.47364 0.33294 0.30380 11.00000 -1.20000
AFIX 0
AFIX 65
C119 1 0.52129 0.34473 0.26874 11.00000 0.14866 0.18305 0.12721 =
-0.01332 0.03245 -0.01800
C126 1 0.56046 0.36791 0.26242 11.00000 0.13851 0.16701 0.13289 =
-0.01976 0.02971 -0.00613
C171 1 0.57874 0.39514 0.28684 11.00000 0.15466 0.18005 0.13187 =
-0.01805 0.03482 -0.02578
C271 1 0.55786 0.39920 0.31759 11.00000 0.17689 0.20790 0.15048 =
-0.03287 0.04918 -0.03633
AFIX 43
H271 6 0.57035 0.41781 0.33428 11.00000 -1.20000
AFIX 0
AFIX 65
C303 1 0.51869 0.37601 0.32391 11.00000 0.18286 0.23320 0.15983 =
-0.03283 0.06494 -0.04405
AFIX 0
C304 1 0.46302 0.35223 0.36658 11.00000 0.26424 0.32841 0.18540 =
-0.05446 0.09986 -0.09753
AFIX 33
H30A 6 0.44352 0.35152 0.34702 11.00000 -1.50000
H30B 6 0.47383 0.32210 0.37132 11.00000 -1.50000
H30C 6 0.44633 0.36343 0.38581 11.00000 -1.50000
AFIX 0
C305 1 1.16093 0.35132 0.09307 11.00000 0.14890 0.22171 0.19543 =
0.03623 0.03832 0.00080
AFIX 33
H30D 6 1.15709 0.32068 0.10095 11.00000 -1.50000
H30E 6 1.18793 0.35325 0.07943 11.00000 -1.50000
H30F 6 1.16364 0.37138 0.11221 11.00000 -1.50000
AFIX 0
C306 1 0.53316 0.38438 0.38973 11.00000 0.26239 0.34947 0.18615 =
-0.03674 0.06206 -0.06927
AFIX 33
H30G 6 0.55733 0.40453 0.38325 11.00000 -1.50000
H30H 6 0.51797 0.39630 0.40941 11.00000 -1.50000
H30I 6 0.54546 0.35496 0.39491 11.00000 -1.50000
AFIX 0
AFIX 66
C307 1 0.52018 0.28384 0.22562 11.00000 0.14170 0.15791 0.09491 =
0.00675 0.02033 -0.01982
AFIX 43
H307 6 0.55176 0.28211 0.22601 11.00000 -1.20000
AFIX 0
AFIX 65
C122 1 0.49828 0.31503 0.24527 11.00000 0.14524 0.17009 0.11898 =
-0.00635 0.02606 -0.01270
C151 1 0.45207 0.31756 0.24471 11.00000 0.15235 0.18277 0.12201 =
-0.01088 0.01827 -0.01169
AFIX 43
H151 6 0.43710 0.33888 0.25815 11.00000 -1.20000
AFIX 0
AFIX 65
C170 1 0.42775 0.28891 0.22450 11.00000 0.14674 0.17621 0.11389 =

-0.00281 0.00958 -0.00210
AFIX 43
H170 6 0.39617 0.29065 0.22412 11.00000 -1.20000
AFIX 0
AFIX 65
C273 1 0.44965 0.25773 0.20485 11.00000 0.14516 0.16113 0.12396 =
-0.00177 0.00295 -0.00759
N076 2 0.49586 0.25519 0.20541 11.00000 0.14771 0.15543 0.11043 =
0.00085 0.01072 -0.02673
AFIX 0
C310 1 0.63570 0.45053 0.29863 11.00000 0.14212 0.14099 0.11090 =
-0.00568 0.02068 0.00019
AFIX 43
H310 6 0.62340 0.45889 0.31951 11.00000 -1.20000
AFIX 0
C311 1 0.82301 0.48434 0.11292 11.00000 0.15876 0.12785 0.14839 =
0.01741 0.03964 0.00183
AFIX 66
C130 1 0.44934 0.08585 0.13230 11.00000 0.15181 0.15152 0.14666 =
0.02191 -0.06035 -0.01634
C138 1 0.40506 0.08024 0.12315 11.00000 0.14623 0.17157 0.17792 =
0.01899 -0.05699 -0.01620
AFIX 43
H138 6 0.39570 0.05327 0.11296 11.00000 -1.20000
AFIX 0
AFIX 65
C227 1 0.37445 0.11411 0.12891 11.00000 0.14801 0.18305 0.19550 =
0.01024 -0.04469 -0.01918
C312 1 0.38814 0.15357 0.14382 11.00000 0.13263 0.16459 0.16981 =
0.01379 -0.03775 -0.00696
AFIX 43
H312 6 0.36722 0.17672 0.14776 11.00000 -1.20000
AFIX 0
AFIX 65
C313 1 0.43243 0.15918 0.15298 11.00000 0.13216 0.14869 0.15349 =
0.02374 -0.03668 -0.01958
C314 1 0.46303 0.12532 0.14721 11.00000 0.14262 0.14540 0.16187 =
0.01189 -0.03711 -0.02202
AFIX 0
C315 1 0.96073 0.49536 0.12793 11.00000 0.16447 0.13777 0.16762 =
0.00424 0.04021 -0.00763
AFIX 43
H315 6 0.96423 0.52505 0.12019 11.00000 -1.20000
AFIX 0
PART 1
C324 1 0.77534 0.29669 0.16369 21.00000 0.19407 0.17829 0.20930 =
0.01177 0.02351 0.01640
AFIX 23
H32A 6 0.76165 0.28757 0.14232 21.00000 -1.20000
H32B 6 0.79891 0.31869 0.15856 21.00000 -1.20000
AFIX 0
C325 1 0.79688 0.25615 0.17951 21.00000 0.18453 0.18378 0.21638 =
0.01632 0.02126 0.01556
AFIX 23
H32C 6 0.77608 0.24358 0.19617 21.00000 -1.20000

H32D 6 0.82403 0.26579 0.19149 21.00000 -1.20000
AFIX 0

C254 1 0.77136 0.35680 0.07564 21.00000 0.18866 0.25063 0.20965 =
-0.01523 -0.00564 -0.00501

AFIX 13

H254 6 0.79875 0.36563 0.08818 21.00000 -1.20000

AFIX 0

C255 1 0.78275 0.31776 0.05357 21.00000 0.18006 0.26475 0.20047 =
-0.01827 -0.00565 -0.00417

AFIX 13

H255 6 0.79468 0.29573 0.06998 21.00000 -1.20000

AFIX 0

CI25 5 0.73340 0.29122 0.03978 21.00000 0.19174 0.25332 0.20016 =
-0.03235 0.00777 -0.02159

CI26 5 0.82997 0.32814 0.02817 21.00000 0.16420 0.34047 0.18361 =
-0.02410 -0.00402 0.01695

CI27 5 0.76138 0.39703 0.04442 21.00000 0.19989 0.21049 0.23070 =
-0.03438 -0.04051 -0.00841

CI28 5 0.73171 0.33609 0.10454 21.00000 0.18997 0.23021 0.22316 =
0.00680 0.01134 0.01516

C257 1 0.67004 0.21858 0.12894 21.00000 0.15239 0.13532 0.15810 =
0.01368 0.03228 -0.01169

AFIX 23

H25G 6 0.66858 0.19691 0.11039 21.00000 -1.20000

H25H 6 0.69692 0.23688 0.12488 21.00000 -1.20000

AFIX 0

C259 1 0.68044 0.19095 0.16021 21.00000 0.12772 0.11853 0.15291 =
0.01392 0.02592 -0.01774

AFIX 23

H25I 6 0.69195 0.21169 0.17733 21.00000 -1.20000

H25J 6 0.65185 0.17906 0.16867 21.00000 -1.20000

AFIX 0

C260 1 0.71308 0.15188 0.15799 21.00000 0.12681 0.11427 0.15050 =
0.02197 0.01634 -0.01695

AFIX 23

H26J 6 0.74081 0.16264 0.14736 21.00000 -1.20000

H26K 6 0.70006 0.12903 0.14313 21.00000 -1.20000

AFIX 0

PART 0

PART 2

C261 1 0.77236 0.18377 0.15478 -21.00000 0.15079 0.14228 0.17775 =
0.02541 0.01596 0.00765

AFIX 23

H26L 6 0.74646 0.19635 0.14276 -21.00000 -1.20000

H26M 6 0.78232 0.15717 0.14227 -21.00000 -1.20000

AFIX 0

C247 1 0.75691 0.16888 0.18866 -21.00000 0.13933 0.12358 0.16317 =
0.01549 0.00843 -0.00303

AFIX 23

H24V 6 0.74222 0.19459 0.19950 -21.00000 -1.20000

H24W 6 0.78370 0.16173 0.20200 -21.00000 -1.20000

AFIX 0

C321 1 0.69123 0.31016 0.12666 -21.00000 0.18071 0.17066 0.18429 =

0.01503 0.03553 -0.00443
AFIX 23
H32E 6 0.67084 0.33503 0.13210 -21.00000 -1.20000
H32F 6 0.69813 0.31086 0.10256 -21.00000 -1.20000
AFIX 0
C322 1 0.67186 0.26642 0.13666 -21.00000 0.17202 0.15590 0.17344 =
0.00886 0.03247 -0.00492
AFIX 23
H32G 6 0.69407 0.24329 0.13093 -21.00000 -1.20000
H32H 6 0.66934 0.26664 0.16124 -21.00000 -1.20000
AFIX 0
C323 1 0.73381 0.31223 0.14772 -21.00000 0.18809 0.17229 0.19764 =
0.00993 0.02509 0.01076
AFIX 23
H32I 6 0.74937 0.28382 0.14313 -21.00000 -1.20000
H32J 6 0.75213 0.33572 0.13715 -21.00000 -1.20000
AFIX 0
HKLF 4

END

Updated SHELX file for second structure:

TITL
CELL 0.71073 29.8662 29.8662 40.7414 90 90 90
ZERR 4 0.0012 0.0012 0.002 0 0 0
LATT 2
SYMM -X,0.5-Y,+Z
SYMM 0.75-Y,0.25+X,0.25+Z
SYMM 0.25+Y,0.25-X,0.25+Z
SFAC C H N O Zn Cl
UNIT 896 896 96 128 48 48
DFIX 1.51 C73 C74
DFIX 1.77 C74 Cl1
DFIX 1.77 C74 Cl3
DFIX 1.77 C73 Cl0
DFIX 1.77 C73 Cl2
DFIX 1.51 0.005 C69 C70 C77 C70 C77 C67 C69 C62 C62 C61
DFIX 1.54 C48 C76A C48 C56A C48 C60A
DFIX 2.5 C76A C60A C60A C56A C56A C76A
DFIX 1.54 C48 C76B C48 C56B C48 C60B
DFIX 2.5 C76B C60B C60B C56B C56B C76B
DFIX 1.54 C45 C50A C45 C57A C45 C63A
DFIX 2.5 C50A C57A C57A C63A C63A C50A
DFIX 1.54 C45 C50B C45 C57B C45 C63B
DFIX 2.5 C50B C57B C57B C63B C63B C50B
DFIX 2.55 0.01 C67 C70 C77 C69 C70 C62
DFIX 1.67 C48 C36
DFIX 1.54 C47 C72A C47 C75A C47 C53A
DFIX 2.5 C53A C75A C75A C72A C72A C53A
SADI C47 C0AA C75A C47 C47 C35 C53A C47 C5 C47 C72A C47
SADI C0AA C5 C5 C35 C35 C0AA C72A C75A C75A C53A C72A C53A
SIMU 0.01 0.02 2
RIGU

L.S. 12 0 287

PLAN 40
SIZE 0.05 0.11 0.11
TEMP -153
ABIN
BOND \$H
CONF
fmap 2
acta
MERG 2
OMIT 2 2 2
OMIT 2 4 4
OMIT -1 7 2
OMIT 1 4 11
OMIT 2 3 3
OMIT 0 5 1
OMIT 0 1 3
OMIT -2 3 1
WGHT 0.1784 348.21991
FVAR 0.02404 0.64637 0.62304 0.72139

Zn1 5 0.09794 0.06017 0.07380 11.00000 0.06120 0.06827 0.05508 =
-0.00628 -0.00380 0.00366
Zn2 5 0.22344 0.27304 0.11911 11.00000 0.05480 0.06547 0.05729 =
-0.00076 -0.01440 -0.00789
Zn3 5 -0.14671 0.04174 0.07543 11.00000 0.08783 0.05952 0.05869 =
-0.01554 0.01617 -0.01628
O4 4 0.23233 0.33874 0.13927 11.00000 0.05789 0.06885 0.04510 =
-0.00229 -0.00790 -0.01252
O2 4 0.15850 0.09411 0.07984 11.00000 0.05238 0.07582 0.05753 =
-0.01106 -0.00485 0.00558
N1 3 0.28983 0.27139 0.13524 11.00000 0.05280 0.07895 0.05347 =
-0.00654 -0.00666 -0.00419
N3 3 0.04678 0.00788 0.04333 11.00000 0.07168 0.07192 0.04490 =
-0.01224 -0.00959 -0.00202
O3 4 -0.08831 0.00982 0.06093 11.00000 0.09112 0.06065 0.06772 =
-0.01880 0.01398 -0.01492
N4 3 0.24744 0.20334 0.10700 11.00000 0.05022 0.07957 0.06327 =
-0.00816 0.00114 -0.00726
N7 3 -0.20861 0.03726 0.10527 11.00000 0.08842 0.06205 0.06229 =
-0.01610 0.01744 -0.02096
N5 3 0.13543 0.03105 0.03786 11.00000 0.07013 0.07043 0.05536 =
-0.00908 0.00507 -0.00531
N6 3 -0.16409 -0.02720 0.07827 11.00000 0.08576 0.06194 0.08875 =
-0.01962 0.02006 -0.02375
C7 1 -0.01376 -0.04276 0.03121 11.00000 0.07911 0.06229 0.05398 =
-0.00687 -0.00381 -0.00501
C8 1 -0.26557 0.06853 0.14019 11.00000 0.08116 0.06698 0.07250 =
-0.01105 0.02512 -0.01964
C9 1 0.25142 0.12811 0.08250 11.00000 0.05175 0.08709 0.07513 =
-0.01972 -0.00432 -0.00815
C10 1 0.30655 0.31341 0.14628 11.00000 0.05313 0.08242 0.04910 =
-0.00322 -0.00570 -0.00939
C11 1 0.05781 -0.05318 0.00655 11.00000 0.07979 0.07398 0.05479 =
-0.01659 0.00533 -0.00980
AFIX 43

H11 2 0.07672 -0.06924 -0.00805 11.00000 -1.20000
AFIX 0
C12 1 0.07449 -0.01661 0.02390 11.00000 0.07106 0.07258 0.04802 =
-0.01162 0.00051 -0.00371
C13 1 0.12039 -0.00193 0.02015 11.00000 0.07278 0.07597 0.05647 =
-0.00955 -0.00015 -0.00496
AFIX 43
H13 2 0.13949 -0.01635 0.00480 11.00000 -1.20000
AFIX 0
C14 1 0.27567 0.34769 0.14500 11.00000 0.05527 0.07402 0.05125 =
-0.00004 -0.01148 -0.01289
C15 1 0.17967 0.04564 0.03630 11.00000 0.06943 0.07458 0.06401 =
-0.00953 0.00339 -0.00130
C16 1 0.29067 0.19665 0.11841 11.00000 0.05242 0.07769 0.06751 =
-0.01510 -0.00131 -0.00303
C17 1 -0.22969 0.07206 0.11720 11.00000 0.08214 0.06837 0.06509 =
-0.01610 0.01962 -0.02019
AFIX 43
H17 2 -0.22068 0.10105 0.11023 11.00000 -1.20000
AFIX 0
C18 1 0.01514 -0.06566 0.01046 11.00000 0.08104 0.06807 0.05421 =
-0.01396 -0.00332 -0.01069
AFIX 43
H18 2 0.00430 -0.09091 -0.00129 11.00000 -1.20000
AFIX 0
C19 1 0.00441 -0.00503 0.04719 11.00000 0.07422 0.07088 0.04346 =
-0.01017 -0.00285 -0.00510
AFIX 43
H19 2 -0.01438 0.01193 0.06131 11.00000 -1.20000
AFIX 0
C20 1 0.23387 0.09472 0.05959 11.00000 0.05607 0.08832 0.07061 =
-0.02103 0.00029 -0.00417
C21 1 0.18954 0.08040 0.06016 11.00000 0.05541 0.07704 0.06506 =
-0.01337 -0.00158 0.00091
C22 1 0.31418 0.15866 0.11384 11.00000 0.05390 0.08370 0.08531 =
-0.02059 -0.00448 -0.00172
AFIX 43
H22 2 0.34323 0.15500 0.12296 11.00000 -1.20000
AFIX 0
C24 1 0.22897 0.17023 0.08918 11.00000 0.05017 0.08072 0.06442 =
-0.00739 -0.00514 -0.01010
AFIX 43
H24 2 0.19977 0.17464 0.08053 11.00000 -1.20000
AFIX 0
C26 1 0.35029 0.31949 0.15664 11.00000 0.05838 0.09935 0.07342 =
-0.00190 -0.01757 -0.01197
AFIX 43
H26 2 0.37031 0.29480 0.15749 11.00000 -1.20000
AFIX 0
C28 1 -0.09311 -0.03115 0.05396 11.00000 0.08711 0.06060 0.06706 =
-0.01168 0.01065 -0.01047
C29 1 0.29384 0.12420 0.09484 11.00000 0.05421 0.08801 0.09135 =
-0.02605 -0.00223 0.00234
AFIX 43
H29 2 0.31028 0.09759 0.09059 11.00000 -1.20000

AFIX 0

C30 1 -0.13479 -0.05414 0.06063 11.00000 0.09167 0.05994 0.08160 =
-0.00833 0.01784 -0.00985

C31 1 -0.05985 -0.05939 0.03800 11.00000 0.08660 0.05935 0.05867 =
-0.00824 0.00249 -0.00576

C32 1 0.29144 0.39158 0.14904 11.00000 0.06401 0.08079 0.06877 =
-0.00235 -0.01858 -0.01433

C33 1 0.31080 0.23424 0.13390 11.00000 0.05277 0.08238 0.06237 =
-0.01209 -0.00932 -0.00319

AFIX 43

H33 2 0.33984 0.23169 0.14324 11.00000 -1.20000

AFIX 0

C34 1 -0.07120 -0.10272 0.02998 11.00000 0.09116 0.05599 0.06567 =
-0.00963 0.00630 -0.00650

AFIX 43

H34 2 -0.04880 -0.12040 0.01968 11.00

Responses to the reviewer's comments

Following the comments from the reviewer 1, we have revised our manuscript. *The reviewer's comments are cited in italics and in blue letters.* Word files of the manuscript and the supplementary information with "Track Changes" records (from the 2nd revision) have been separately uploaded.

To reviewer 1

We would like to thank the present reviewer for helpful attachments and comments in revising the crystallography. We have revised the crystal structure analysis after carefully considering the attachments.

Additional corrections are still needed for these structures.

Quote "The hydrogens at the methylene groups belonging to the "branched part" of the central dicarboxylates were not modelled due to the limitation of the SHELX program." - this is not true. These hydrogens can be modelled with SHELXL.

Quote "We tried to replace SIMU RIGU and ISOR restraints. The trials were not successful. So, we still use many restraints." - following the review is a copy of the converged structure where global RIGU and SIMU restraints have been applied, and the hydrogen atoms on the methylene atoms have been correctly modelled. Note that this refinement method removes the five B-level alerts relating to low Ueq values, and results in more sensible anisotropic thermal parameters over the whole model. It also removes the need for the XNPD restraint, and removes the need to DAMP the structure refinement.

Now we have understood what we should do from the attachment. We did the refinement based on the attachment. We added DISP commands for Zn and Cl atoms. Many alerts as well as many restraints were successfully removed by this refinement method. We would really like to thank the present reviewer.

Although the _vrf's now have more sensible comments, they are now no longer syntactically correct. For example, "_vrf_PLAT242_shelx" has become "_vrf_PLAT242". Omitting the _shelx identifier means the _vrf is no longer associated with the _shelx data block, and the _vrf no longer appears in the checkCIF report. This should be corrected for both structures.

We corrected _vrf's. The _vrf's now appear in the checkCIF report.

Minor correction -

_diffn_radiation_monochromator 'Synchrotron' - should read 'Si double crystal monochromator'

We changed 'Synchrotron' to 'Si double crystal monochromator'

Regarding the second structure:

Quote "We changed an atomic coordinate of O10 to be connected to C66. The two parts of the outer ligand has been joined together. We also modelled the rotational isorder of two of the t-butyl groups." The reviewer must apologize, there is also rotational disorder of the third t-butyl group, which needs to be modelled.

For this structure, again, refining with global RIGU and SIMU restraints (the SIMU slightly tightened to "SIMU 0.01 0.02 2") results in overall better quality thermal parameters, and removes the need entirely for any DAMP and XNPD restraints, and EADP constraints. An updated SHELX file has been appended to the report, with thermal paramter constraints and disorder modelled.

As with the first structure, now we have understood what we should do from the attachment. We did the final refinement based on the attachment. We additionally omitted one reflection using OMIT instruction. Many alerts as well as many restraints were successfully removed by this refinement method. We really appreciate the help from the reviewer.

Additional points from the manuscript:

Page 19, line 16: "The diffraction intensity data up to 1.00 Å was measured..." - in the structure refinement the applied data limit was 0.9 Å. It is also implied in the CIF that data was collected to a higher angle, however was cut down during structure refinement do to poor quality of diffraction at higher angles. The manuscript should be updated to reflect this.

Thank you for your pointer. We changed the manuscript as follows.

· Page 17, line 16

(Original) The diffraction intensity data up to 1.00 Å was measured

(Revised) The diffraction intensity data was measured

· **Page 17, line 19 and page 19, line 1**

We added the sentence below.

(Revised) The diffraction data up to 0.9 Å was used for the structure refinement.

Page 19, line 18: "The structure was determined and refined using SHELXL-2014" - SHELXL-2016 is the version used to refine the structure, and the reference should be updated to the 2015 paper (<http://dx.doi.org/10.1107/S2053229614024218>). SHELXL is not a structure solution program. The program used to solve the structure (SIR92) should be included and appropriately referenced.

Page 20, line 23: "Obtained data were processed using a Bruker APEX2 and refined using SHELXL-2014." - The SHELXL version should be correct, and the reference updated. The program used to solve the structure (SHELXS) should be mentioned and appropriately reference (in this case the 2008 paper is the correct reference - <https://doi.org/10.1107/S0108767307043930>).

We appreciate your comments. The references have been corrected according to your suggestions. Please see the revised manuscript and cif files.

According to the changes described above, Fig. 3 and Fig. 4b,c in the manuscript, Supplementary Figs. S14, S22, and S23 have been redrawn with the revised structures. Descriptions in the Methods section have been also revised.

REVIEWERS' COMMENTS:

Reviewer #1 (Remarks to the Author):

I am now satisfied with the CIFs and changes to the manuscript, and can recommend them for publication.